# DREAMLLM: SYNERGISTIC MULTIMODAL COMPREHENSION AND CREATION

**Runpei Dong** [12⋆†]    **Chunrui Han** [3⋆]    **Yuang Peng** [4†]    **Zekun Qi** [12†]    **Zheng Ge** [3]
**Jinrong Yang** [5†]    **Liang Zhao** [3]    **Jianjian Sun** [3]    **Hongyu Zhou** [3]    **Haoran Wei** [3]
**Xiangwen Kong** [3]    **Xiangyu Zhang** [3‡]    **Kaisheng Ma** [4¶]    **Li Yi** [467¶‡]

[1]Xi'an Jiaotong University    [2]Institute for Interdisciplinary Information Core Technology (IIISCT)
[3]MEGVII Technology    [4]Tsinghua University    [5]HUST
[6]Shanghai Artificial Intelligence Laboratory    [7]Shanghai Qi Zhi Institute

## ABSTRACT

This paper presents DREAMLLM, a learning framework that first achieves versatile Multimodal Large Language Models (MLLMs) empowered with frequently over-looked synergy between multimodal comprehension and creation. DREAMLLM operates on two fundamental principles. The first focuses on the generative modeling of both language and image posteriors by direct sampling in the raw multimodal space. This approach circumvents the limitations and information loss inherent to external feature extractors like CLIP, and a more thorough multimodal understanding is obtained. Second, DREAMLLM fosters the generation of raw, interleaved documents, modeling both text and image contents, along with unstructured layouts. This allows DREAMLLM to learn all conditional, marginal, and joint multimodal distributions effectively. As a result, DREAMLLM is the first MLLM capable of generating free-form interleaved content. Comprehensive experiments highlight DREAMLLM's superior performance as a zero-shot multimodal generalist, reaping from the enhanced learning synergy. Project page: `dreamllm.github.io`.

## 1    INTRODUCTION

> "What I cannot create, I do not understand."
>
> *Richard P. Feynman, on his blackboard at the time of his death, 1988*

Content *comprehension* and *creation* in multimodality are crucial and among the ultimate courses of machine intelligence (Sternberg, 1985; Legg & Hutter, 2007). To this end, Multimodal Large Language Models (MLLMs) (Alayrac et al., 2022; Hao et al., 2022; Huang et al., 2023) have emerged as extensions of the successful GPT-style Large Language Models (LLMs) (Brown et al., 2020; Zhang et al., 2022; OpenAI, 2022; 2023a;b; Chen et al., 2023b; Touvron et al., 2023a;b) into visual realm. Recognized as foundation models (Bommasani et al., 2021), MLLMs have achieved unprecedented progress in multimodal comprehension capabilities. These advanced models typically enhance LLMs by incorporating images as multimodal inputs, such as CLIP features (Radford et al., 2021), to facilitate language-output multimodal comprehension. Their aim is to capture multimodal conditional or marginal distributions via a *language posterior*. However, multimodal creation, which involves generating images, texts, or both, necessitates a *universal* generative model that simultaneously learns language and image posteriors—currently underexplored.

Until very recently, some concurrent works have shown success in conditional image generation using MLLMs (Koh et al., 2023; Sun et al., 2023b). As depicted in Fig. 1, these methods compel MLLMs to produce either discrete or continuous conditional embeddings that explicitly align with a pretrained CLIP encoder, which could later be used by a pretrained Stable Diffusion (SD) (Rombach et al., 2022) model for image generation. However, due to an inherent modality gap (Liang et al., 2022), CLIP

---

⋆Equal contribution. †Work partially done during the internship at IIISCT and MEGVII. ‡Project leaders. ¶Corresponding authors.

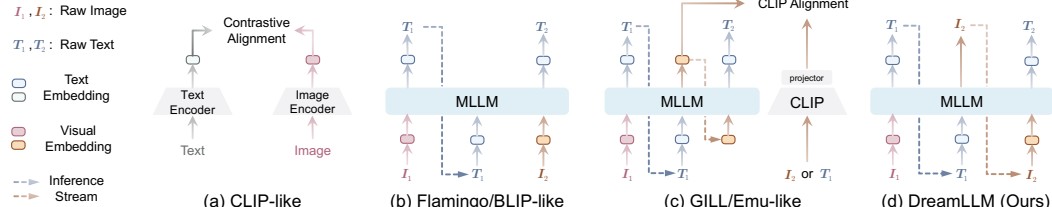

Figure 1: **Conceptual comparison** of vision-language (VL) foundation models. (a) CLIP-like models (Radford et al., 2021; Yu et al., 2022a; Li et al., 2023e) take advantage of two towers that explicitly align VL representations. (b) Flamingo/BLIP-like models (Alayrac et al., 2022; Li et al., 2022; 2023d; Huang et al., 2023) encode VL representations into a unified manifold space using a singular MLLM. However, these models lack full autoregressivity, as they only output language. (c) Concurrent MLLMs (Koh et al., 2023; Sun et al., 2023b) align visual outputs with CLIP representations, but this alignment occurs in an intermediate space, not a raw data space. Consequently, models such as Emu necessitate a second-stage fine-tuning of Stable Diffusion (Rombach et al., 2022) for raw image generation. These models also fall short in generating raw interleaved documents. (d) Our DREAMLLM, instead, generates raw language and image inputs in a unified auto-regressive manner, inherently enabling interleaved generation. Only non-autoregressive generation loss is noted.

semantics focus predominantly on *modality-shared* information, often overlooking *modality-specific* knowledge that could enhance multimodal comprehension. Consequently, these studies have *not* fully realized the potential learning synergy between multimodal creation and comprehension, have shown only *marginal* improvements in creativity, and remain *deficient* in multimodal comprehension.

In this work, we introduce DREAMLLM, universally learning image and text posteriors with expected creation & comprehension synergy, based on the following two *de-facto* designing principles:

i. **Generate Everything as It Is** Different from existing works that generate intermediate image representations like CLIP embeddings during training, DREAMLLM not only takes all modalities raw data as inputs but also as outputs in a truly end-to-end fashion (*i.e.*, outputs are *identical* to inputs, see Fig. 1). The challenge lies in enabling MLLMs to learn the image posterior without compromising their comprehension capabilities. To address this, we introduce *dream queries*, a set of learnable embeddings that encapsulate the semantics encoded by MLLMs. This approach avoids altering the output space of MLLMs. Raw images are then decoded by the SD image decoder conditioned on these semantics. In this fashion, the pretrained SD acts as the *score function* (Ho et al., 2020). The *image posterior* is thus modeled by direct sampling in the pixel space, facilitated by *score distillation* (van den Oord et al., 2018; Poole et al., 2023).

ii. **Interleaved Generative Pre-Training ($\mathcal{I}$-GPT)** DREAMLLM is trained to generate interleaved multimodal corpora from the internet (Zhu et al., 2023b), both *encoding* and *decoding* interleaved image-text multimodal inputs. Unlike encoding multimodal inputs as in existing methods, decoding interleaved multimodal outputs is challenging due to the complex interleaving layout structures and the long-context requirement of images. Our approach tackles the interleaved layout learning using a unique <dream> token that predicts the placement of images within texts. Harnessing DREAMLLM's causal nature, all contents are generated with history multimodal contexts of any length. This *interleaved generative pretraining ($\mathcal{I}$-GPT)* inherently forms all joint, marginal, and conditional distributions of images and texts in the document, leading to a learning *synergy* that grounds DREAMLLM's comprehension in creation and vice versa.

Extensive experiments across various vision-language comprehension, content creation, and language-only tasks demonstrate DREAMLLM's superior performance as a *zero-shot multimodal generalist*. For instance, DREAMLLM-7B achieves an 8.46 FID on MS-COCO and sets a new standard with 49.1/35.9 scores on MMBench and MM-Vet evaluations, respectively. Moreover, we delve into the learning synergy between comprehension and creation, revealing decent in-context generation capabilities. With $\mathcal{I}$-GPT pretraining, DREAMLLM generates interleaved documents following human prompts after supervised fine-tuning on instruction-following data curated with GPT-4. To our knowledge, this work is the first to enable MLLMs to create free-form interleaved content with a learning synergy on both sides. As a foundational learning framework, DREAMLLM is adaptable across all modalities, laying a promising foundation for future multimodal learning research.

## 2 BACKGROUND & PROBLEM STATEMENT

**Autoregressive Generative Modeling** Given the joint probability distribution $p_\theta(\mathbf{w})$ over a sequence $\mathbf{w} = \{\mathbf{w}_t\}_{t=1}^T$ with length $T$, the canonical causal generation (Mikolov et al., 2010; Radford et al., 2018; 2019) of every token $\mathbf{w}_t$ by a $\theta$-parameterized language model $\mathcal{F}$ is modeled as $p_\theta(\mathbf{w}) = \prod_{t=1}^T p_\theta(\mathbf{w}_t|\mathbf{w}_{<t})$. For multimodal comprehension, the sequence could contain $K$ ordered images $\boldsymbol{I} = \{I_k\}_{k=1}^K$ interleaved with words. The $k$-th image is processed as patch embeddings with visual encoders $\mathcal{H}_\phi(\cdot)$ like CLIP, which will then be encoded by a projector $\mathcal{M}_\zeta$ (*e.g.*, a linear layer (Huang et al., 2023) or DETR- (Carion et al., 2020)/Perceiver-like (Jaegle et al., 2021) Resampler (Alayrac et al., 2022)) into $L$-length visual embeddings $\boldsymbol{V}_k = \{\mathbf{v}_\ell\}_{\ell=1}^L$. Let $K(t)$ be the image number before the $t$-th word token. The maximum likelihood estimation (MLE) is to minimize

$$\mathcal{L}_{\text{MLLM}}(\Theta = \{\theta, \zeta\}, \mathbf{w}, \boldsymbol{I}) := -\mathbb{E}_t\left[\log p_\Theta(\mathbf{w}_t|\mathbf{w}_{<t}, \boldsymbol{V}_{<K(t)})\right], \quad \boldsymbol{V}_{K(t)} = \mathcal{M}_\zeta \circ \mathcal{H}_\phi(I_{K(t)}). \quad (1)$$

**Diffusion Models** Diffusion Models (DMs) (Sohl-Dickstein et al., 2015; Ho et al., 2020) are probabilistic generative models that learn the latent structure of data $\mathbf{z} = \{\mathbf{z}_t\}_{t=1}^T$ through continuous-$T$-timestamps information diffusion. DMs involve a forward or diffusion process $q$ that smoothly converts data to Gaussian noise. Given the initial datapoint $\mathbf{z}_1 \sim q(\mathbf{z}_1)$ and diffusion rate $\beta_t := 1-\alpha_t$, this process can be defined as a marginal distribution $q(\mathbf{z}_t|\mathbf{z}_1) := \mathcal{N}(\sqrt{\alpha_t}\mathbf{z}_1, \beta_t\mathbf{I})$, and the perturbed data distribution is $q(\mathbf{z}_t) := \int q(\mathbf{z}_t|\mathbf{z})q(\mathbf{z})d\mathbf{z}$ by integrating out data density $q(\mathbf{z})$. A reversed denoising probability flow $p$ is used for generating data from noise $\mathbf{z}_T \sim \mathcal{N}(\mathbf{0}, \mathbf{I})$ as a Markov Chain with transition approximated by a Gaussian model $p_\xi(\mathbf{z}_{t-1}|\mathbf{z}_t) := \mathcal{N}(\boldsymbol{\mu}_\xi(\mathbf{z}_t), \sigma_t^2\mathbf{I})$, which relates to an optimal MSE denoiser since $q(\mathbf{z}_{t-1}|\mathbf{z}_t, \mathbf{z}_1)$ is Gaussian with enough timestamps (Feller, 1949; Sohl-Dickstein et al., 2015). Ho et al. (2020) show that the optimization with the evidence lower bound (ELBO) can be simplified by training a denoising U-Net $\boldsymbol{\epsilon}_\xi(\mathbf{z}_t, t)$ parameterized with $\xi$ that estimates the conditional expectation $\mathbb{E}[\boldsymbol{\epsilon} \sim \mathcal{N}(\mathbf{0}, \mathbf{I})|\mathbf{z}_t]$ (Bao et al., 2022). Let $\mathcal{C}$ be the conditional embeddings, and the perturbed data $\mathbf{z}_t = \sqrt{\overline{\alpha}_t}\mathbf{z}_1 + \sqrt{1-\overline{\alpha}_t}\boldsymbol{\epsilon}$, the minimization objective is

$$\mathcal{L}_{\text{DM}}(\xi, \mathbf{z}) := \mathbb{E}_{t\sim\mathcal{U}(0,1),\boldsymbol{\epsilon}\sim\mathcal{N}(\mathbf{0},\mathbf{I})}\left[\|\boldsymbol{\epsilon}_\xi(\mathbf{z}_t;\mathcal{C},t) - \boldsymbol{\epsilon}\|^2\right]. \quad (2)$$

Since $\boldsymbol{\epsilon}_\xi(\mathbf{z}_t;t) = -\sigma_t s_\xi(\mathbf{z}_t;t)$ as derived from Tweedie's (Efron, 2011; Luo, 2022), it is equivalent to denoising score matching of $\boldsymbol{\nabla}_{\mathbf{z}_t} \log p_\xi(\mathbf{z}_t)$ (Hyvärinen, 2005; Vincent, 2011), thus DMs are also called *score-function* based generative models (Song & Ermon, 2019; 2020; Song et al., 2021; 2023).

### 2.1 HOW CAN WE USE MLLMs FOR DIFFUSION SYNTHESIS THAT SYNERGIZES BOTH SIDES?

Multimodal signals typically exhibit *modality-specific* information that has distinct structure but *complementary* semantics (Dong et al., 2023). This complementary property allows us to utilize deep language comprehension to enhance cross-modal image generation (Saharia et al., 2022). However, the potential of multimodal creation to improve comprehension remains largely unexplored.

Existing strategies (Koh et al., 2023; Sun et al., 2023b; Ge et al., 2023) integrate successful Diffusion Models with MLLMs by aligning the semantic spaces of conditional embeddings between CLIP $\mathcal{C}^{\text{CLIP}}$ and MLLMs $\mathcal{C}^{\text{MLLM}}$. The objective is to minimize alignment loss $\mathcal{L}_{\text{align}} = D(\mathcal{M}_\psi \circ \mathcal{C}^{\text{MLLM}}, \mathcal{C}^{\text{CLIP}})$, employing a distance metric $D(\cdot, \cdot)$ and a condition projector $\mathcal{M}_\psi$. However, CLIP models primarily learn *modality-shared* semantics, often overlooking *modality-specific* information due to a modality gap (Liang et al., 2022; Liu et al., 2023f). This explicit alignment with CLIP's intermediate output space may induce more conflicts than synergies, as MLLMs are forced to generate semantically reduced information, deviating from their original output space. To circumvent these issues, we propose alternative learning methodologies (See Fig. 2), which we elaborate in the ensuing sections.

**Learning Objective** Our aim is to leverage MLLMs to model distributions via direct pixel space sampling. Here, the pretrained SD functions as a score metric, distilling the learned data distribution. This approach is similar to Score Distillation Sampling (Poole et al., 2023) (SDS, also known as Score Jacobian Chaining (Wang et al., 2023a)). In this context, image posterior is learned in a DeepDream-like manner (Mordvintsev et al., 2015), using MLLMs' conditional parameterization.

**Conditional Embeddings** Rather than converting the output space of MLLMs to align with CLIP, we propose to *query* MLLMs using learned embeddings. Consequently, MLLMs-enriched semantics serve as diffusion conditioning, and the distribution is implicitly modeled through synthesis sampling.

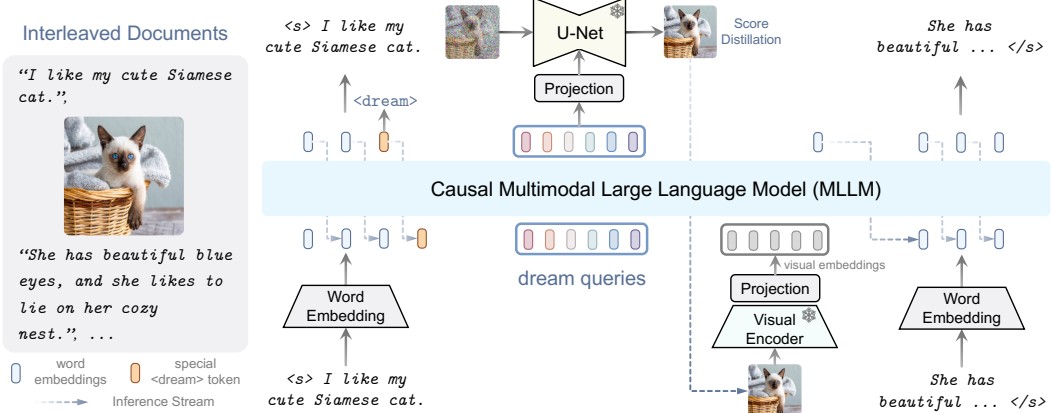

Figure 2: **Overview of of our DREAMLLM framework**. Interleaved documents serve as input, decoded to produce outputs. Both text and images are encoded into sequential, discrete token embeddings for the MLLM input. A special <dream> token predicts where to generate images. Subsequently, a series of *dream queries* are fed into the MLLM, capturing holistic historical semantics. The images are synthesized by the SD image decoder conditioned on queried semantics. The synthesized images are then fed back into the MLLM for subsequent comprehension.

## 3 DREAMLLM

We introduce DREAMLLM, a universal learning framework that facilitates both MLLM's comprehension and creation capabilities. Our DREAMLLM is built with a causal decoder-only LLM $\mathcal{F}_\theta$ as the model foundation, *i.e.*, Vicuna (Chiang et al., 2023) based on LLaMA (Touvron et al., 2023a) trained on ShareGPT (Zheng et al., 2023). We adopt OpenAI's CLIP-Large (Radford et al., 2021) as the visual encoder $\mathcal{H}_\phi$, followed by a linear layer $\mathcal{M}_\zeta$ for visual embedding projection. To synthesize images, we use Stable Diffusion (SD) (Rombach et al., 2022) as the image decoder, and the condition projector $\mathcal{M}_\psi$ is also a linear layer. An overview of the architecture is depicted in Fig. 2.

### 3.1 END-TO-END INTERLEAVED GENERATIVE PRETRAINING ($\mathcal{I}$-GPT)

All natural documents can be regarded as carriers of text-image interleaved information. Text-only, images-only, and text-image pairs data, on the other hand, can be seen as special cases of interleaved corpora with different modality compositions. Thus, it is critical to empower the model with the capability to learn and generate *free-form interleaved documents* that form all possible distributions.

**Interleaved Structure Learning**  To model the interleaved structure, the interleaved sequence is operated by extending a new special <dream> token before images. During training, DREAMLLM is trained to predict this <dream> token that indicates where an image emerges, and the conditional image synthesis is performed afterward, as introduced next. During inference, DREAMLLM will generate an image on its "free will" when this token is predicted.

**Conditional Synthesis through Score Distillation**  To avoid the possible conflicts of CLIP semantics and MLLMs stated in Sec. 2.1, we carefully design a different learning objective and conditional embeddings. Formally, we introduce a series of learnable *dream queries* with length $Q$: $\mathbf{d} = \{\mathbf{d}_q\}_{q=1}^Q$. Considering the $t$-th token is predicted as <dream> token, the conditional embeddings $\mathcal{C}_{K(t)+1}^{\text{DREAMLLM}}$ for the $(K(t)+1)$-th image synthesis can be obtained by causally querying the previous sequences:

$$\mathcal{C}_{K(t)+1}^{\text{DREAMLLM}} := \mathcal{F}_\theta(\mathbf{d}, \mathbf{x}_{<t+1}, \boldsymbol{V}_{<K(t)+1}). \tag{3}$$

Thus, the denoising score matching with latent $\mathbf{z}$ is motivated in the similar formulation to Eq. (2):

$$\mathcal{L}_{\text{DM}}^{\text{DREAMLLM}}(\theta, \mathbf{d}, \zeta, \psi, \mathbf{z}) := \mathbb{E}_{t\sim\mathcal{U}(0,1), \boldsymbol{\epsilon}\sim\mathcal{N}(\mathbf{0},\mathbf{I})} \left[ \|\boldsymbol{\epsilon}_\xi(\mathbf{z}_t; \mathcal{C}^{\text{DREAMLLM}}, t) - \boldsymbol{\epsilon}\|^2 \right], \tag{4}$$

where $\xi$ is not updated since the SD is frozen. Eq. (4) can also be viewed as a generalized formulation of *textual inversion* (Gal et al., 2023), but all condition embeddings are learnable by model-seeking. From the perspective of *score distillation* (van den Oord et al., 2018), the KL divergence defined by conditions and the pre-learned score function is equivalently minimized for distilling (Hinton et al.,

Table 1: **Zero-shot multimodal comprehension evaluation** of image-to-text captioning, general VQA, text-related VQA, and comprehensive benchmarks. * denotes non-zero-shot results for VQA. DREAMLLM-7B* is trained using the SFT data constructed by LLaVA-1.5 (Liu et al., 2023b).

| Method | Captioning | | VQA | | | | Comprehensive | |
|---|---|---|---|---|---|---|---|---|
| | COCO | I2Paragraph | VQAv2 | OKVQA | VizWiz | TextVQA | MMBench | MM-Vet |
| *Comprehension Only MLLMs* | | | | | | | | |
| MetaLM (Hao et al., 2022) | - | - | 41.1 | 11.4 | - | - | - | - |
| Kosmos-1 (Huang et al., 2023) | - | - | 51.0 | - | 29.2 | - | - | - |
| Flamingo-9B (Alayrac et al., 2022) | 79.4 | - | 51.8 | 44.7 | 28.8 | - | - | - |
| OF-9B (Awadalla et al., 2023) | 65.5 | - | 52.7 | 37.8 | 27.5 | 29.1 | 4.6 | 21.8 |
| LLaVA-7B (Liu et al., 2023c) | - | - | - | - | - | 28.9 | 38.7 | 23.8 |
| *MLLMs for Comprehension & Creation* | | | | | | | | |
| CM3Leon-7B* (Yu et al., 2023a) | 61.6 | 10.5 | 47.6 | 23.8 | 37.6 | - | - | - |
| Emu-14B (Sun et al., 2023b) | **117.7** | - | 40.0 | 34.7 | 35.4 | - | - | - |
| DREAMLLM-7B (Ours) | 115.4 | **17.4** | **56.6** | **44.3** | **45.8** | **34.9** | **49.9** | **35.9** |
| DREAMLLM-7B* (Ours) | 103.7 | 8.4 | **72.9** | **52.2** | **49.3** | **41.8** | **58.2** | **36.6** |

2015) learned probability density in conditional image synthesis:

$$\min_{\theta,\mathbf{d},\zeta,\psi} \mathcal{L}_{\text{DM}}^{\text{DREAMLLM}} := \mathbb{E}_{t,\mathcal{C}^{\text{DREAMLLM}}}\left[D_{\text{KL}}\big(q(\mathbf{z}_{t-1}|\mathbf{z}_t,\mathbf{z}_1,\mathcal{C}^{\text{DREAMLLM}}) \parallel p_\xi(\mathbf{z}_{t-1}|\mathbf{z}_t)\big)\right]. \quad (5)$$

**Universal Multimodal Generative Modeling** An interleaved document sequence $\mathbf{x} = \{\mathbf{x}_t\}_{t=1}^T$ contains both words $\mathbf{w} = \{\mathbf{w}_i\}_{i=1}^N$ and images $\boldsymbol{I} = \{I_k\}_{k=1}^K$. The autoregressive nature forms all possible conditional distributions, such as image conditional multimodal comprehension $p(\mathbf{w}|\boldsymbol{I})$ or text-to-image synthesis $p(\boldsymbol{I}|\mathbf{w})$. The images are processed as visual embeddings $\boldsymbol{V}$ for causal comprehension. Assuming that the pretrained SD is an optimal score function, Eq. (5) thus could be viewed as an MLE optimization for the synthesis posterior. Different from Eq. (1), the targeted sequence $\mathbf{x}_t$ now could be both encoded images or words. The objective is thus unified to the MLE of all causally-conditioned posteriors in arbitrary forms:

$$\mathcal{L}_{\text{MLLM}}^{\text{DREAMLLM}}\left(\Theta = \{\theta,\mathbf{d},\zeta,\psi\}, \mathbf{x}\right) := -\mathbb{E}_t\left[\log p_\Theta(\mathbf{x}_t|\mathbf{x}_{<t})\right]. \quad (6)$$

## 3.2 MODEL TRAINING

In this work, we consider a three-stage training procedure. It can be summarized as follows, and the implementation details, like training data, can be found in Table 13 in Appendix C.

  I **Alignment Training** This stage is used to alleviate the gap in multimodality, facilitating the adaptation of multimodal inputs to LLMs. The linear *visual projector*, linear *condition projector*, and learnable *dream embeddings* are pretrained for cross-modal manifold alignment among *frozen* LLMs, visual encoder, and SD. We use approximately 30M image-text pairs data, training both image-to-text comprehension and text-to-image synthesis.

 II **$\mathcal{I}$-GPT Pretraining** Following alignment, the LLM undergoes an *unfrozen* process for $\mathcal{I}$-GPT pretraining (detailed in Sec. 3.1). This critical stage facilitates the learning of joint vision-language distributions via generative modeling. Training incorporates approximately 2M selectively filtered documents from MMC4-Core (Zhu et al., 2023b), adhering to a CLIP score threshold of 0.25. Furthermore, we use 2M paired data samples from LAION400M (Schuhmann et al., 2021), captioned by BLIP (Li et al., 2022) (*i.e.*, BLIP-LAION), to enhance text-to-image training and potentially mitigate the impact of some low-quality noisy images and texts from sMMC4.

III **Supervised Fine-tuning** This stage enables the model to perform general multimodal comprehension and creative tasks following human instructions (Ouyang et al., 2022). We utilize approximately 80K visual instruction tuning data collected by Liu et al.. For instruction-following content creation, GPT-4 is prompted with document summaries or image captions, collecting approximately 20K instruction-following document synthesis from MMC4 (InstructMMC4) and 20K image synthesis data from BLIP captioned LAION400M (Instruct-BLIP-LAION).

## 4 EXPERIMENTS

DREAMLLM is a versatile multimodal generalist that excels at zero-shot or in-context vision-language comprehension and synthesis tasks. In this section, we conduct systematic evaluations for demonstration. See qualitative results in Appendix B and implementation details in Appendix C.

## 4.1 MULTIMODAL COMPREHENSION

Multimodal comprehension enables humans to interact with agents conditioned on both words and visual content. We evaluate the multimodal vision and language capabilities of DREAMLLM across several benchmarks, including image-to-text captioning on COCO (Karpathy & Fei-Fei, 2017) and Image2Paragraph (Krause et al., 2017), general visual question answering (VQA) on VQAv2 (Goyal et al., 2019), OKVQA (Marino et al., 2019), VizWiz (Gurari et al., 2018), and text-related VQA on TextVQA (Singh et al., 2019). Additionally, we conducted a zero-shot evaluation on the recently developed benchmarks of MMBench and MM-Vet to assess the model's performance in complex multimodal tasks. The results are presented in Table 1 (See Table 5, and Table 6 in Appendix A). All metrics and data splits are listed in Table 14 in Appendix C. We find that i) DREAMLLM outperforms other MLLMs across all benchmarks. Notably, DREAMLLM-7B surpasses concurrent MLLMs with image synthesis capabilities by a significant margin, achieving +16.6 higher accuracy on VQAv2 compared to Emu-13B. ii) On comprehensive benchmarks like MMBench and MM-Vet, DREAMLLM achieves state-of-the-art performance against all 7B counterparts. Detailed analysis revealed superior spatial/relation reasoning capabilities in DREAMLLM compared to other MLLMs, likely a result of its image synthesis learning. See *qualitative results and comparisons* on multimodal dialogue in Table 11, Table 12, Fig. 10, Fig. 11, and Fig. 12, in Appendix B.

## 4.2 TEXT-CONDITIONAL IMAGE SYNTHESIS

Text2Image is one of the most commonly used techniques for creative content generation that follows human's fabulous imaginations through free-form languages.

We assess text-conditional image synthesis on the MS-COCO validation set (Lin et al., 2014) and LN-COCO, the COCO subset of Localized Narratives (Pont-Tuset et al., 2020), following prior works (Xu et al., 2018; Yu et al., 2022b). The MS-COCO dataset primarily contains high-level image abstractions with shorter captions, whereas LN-COCO provides more comprehensive image descriptions (Yu et al., 2022b). DREAMLLM samples 8 images per text prompt on MS-COCO by CLIP score ranking, following previous works (Ramesh et al., 2022). On LN-COCO, DREAMLLM samples one image per prompt without CLIP ranking since the text is too long and exceeds the CLIP length limit. Note that Parti samples 16 images per prompt with CoCa (Yu et al., 2022a). Our evaluation metric is the zero-shot Fréchet Inception Distance (FID) (Heusel et al., 2017), the results of

Table 2: **Zero-shot text-to-image generation FID** on MS-COCO LN-COCO. LM denotes *language model* based methods, MG denotes *multimodal generation* methods, and FIG denotes *free-form interleaved generation* methods. $^\dagger$ is fine-tuned SDv2.1 on our state I data. $^*$ denotes retrieval-augmentation (Sheynin et al., 2023). $^\triangleright$ denotes results after stage I alignment training.

| Method | LM | MG | FIG | MS-COCO | LN-COCO |
|---|---|---|---|---|---|
| *Text2Image Specialists* | | | | | |
| Retrieval Result (Yu et al.) | ✗ | ✗ | ✗ | 17.97 | 33.59 |
| DALL-E (Ramesh et al.) | ✗ | ✗ | ✗ | ~28 | - |
| CogView (Ding et al.) | ✗ | ✗ | ✗ | 27.1 | - |
| CogView2 (Ding et al.) | ✗ | ✗ | ✗ | 24.0 | - |
| SDv2.1 (Rombach et al.) | ✗ | ✗ | ✗ | 12.43 | 34.26 |
| SDv2.1$^\dagger$ (Rombach et al.) | ✗ | ✗ | ✗ | 11.91 | 25.35 |
| GLIDE (Nichol et al.) | ✗ | ✗ | ✗ | 12.24 | - |
| Make-A-Scene (Gafni et al.) | ✗ | ✗ | ✗ | 11.84 | - |
| DALL-E 2 (Ramesh et al.) | ✗ | ✗ | ✗ | 10.39 | - |
| Muse-3B (Chang et al.) | ✓ | ✗ | ✗ | 7.88 | - |
| Imagen-3.4B (Saharia et al.) | ✓ | ✗ | ✗ | 7.27 | - |
| Parti-20B (Yu et al.) | ✓ | ✗ | ✗ | **7.23** | **15.97** |
| *Multimodal Large Language Models* | | | | | |
| CM3-13B (Aghajanyan et al.) | ✓ | ✓ | ✗ | 29.56 | - |
| GILL-8B (Koh et al.) | ✓ | ✓ | ✗ | 12.20 | - |
| Emu-13B (Sun et al.) | ✓ | ✓ | ✗ | 11.66 | - |
| CM3Leon-7B$^*$ (Yu et al.) | ✓ | ✓ | ✗ | 10.82 | - |
| DREAMLLM-7B$^\triangleright$ (Ours) | ✓ | ✓ | ✓ | **8.76** | **22.42** |
| DREAMLLM-7B (Ours) | ✓ | ✓ | ✓ | **8.46** | **20.53** |

which are presented in Table 2. We note three key observations: i) Our DREAMLLM shows a significant FID improvement over the StableDiffusion baseline after stage-I alignment, reducing the score by 3.67 and 11.83 on MS-COCO and LN-COCO, respectively. Further, FID improvements of 3.97 and 13.73 are achieved after pretraining and supervised fine-tuning. The substantial improvement on LN-COCO underscores DREAMLLM's superior capability in processing long-context information. ii) When compared to prior specialist models, DREAMLLM delivers competitive results based on the SD image decoder. iii) DREAMLLM consistently outperforms concurrent MLLMs-based image synthesis methods. For instance, DREAMLLM-7B surpasses Emu-13B by a significant 3.20 FID on MS-COCO. See *qualitative results* on text-to-image synthesis in Fig. 13 and Fig. 14 in Appendix B.

## 4.3 MULTIMODAL JOINT CREATION & COMPREHENSION

**Free-form Interleaved Document Creation** Leveraging the interleaved generative modeling from $\mathcal{I}$-GPT, DREAMLLM can now generate *interleaved documents* in a free-form manner. In

👤 *Help me write the introduction and illustrations about Vienna city, show several high-quality photographs about the beautiful scenes and buildings such as Vienna International Opera House.*

⚛ *The Austrian capital, Vienna, has long been known as the City of Dreams, and there is no better place to experience than this romantic city during the holiday season. The city's historic center is surrounded by a collection of beautiful buildings on a square mile. Stroll along the famous Ring Boulevard and admire the architecture. Some of the buildings date back to the 13th century. The Vienna State Opera House is one of the most beautiful buildings. Founded in 1869, the opera house is the* 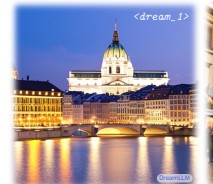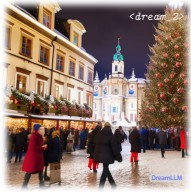 *home of the Vienna State Opera and the annual Vienna Opera Ball. <dream_1> Cosmopolitan flair, an eclectic mix of architectural styles and a rich cultural heritage come together in Vienna. The city is home to a number of museums, palaces and churches. The Sacher Torte and the Wiener Schnitzel are local specialties. A visit to the romantic Christmas Market in Vienna is also a unique experience. <dream_2>*

----------------------------------------------------------------------------------------------------

👤 *Imagine you are a robot agent in the house. There is a mobile phone on the dining table in the dining room, and you are in the living room. Write a short article about how would you walk through the house to grab the mobile phone with your robotic arm step by step. Include several high-quality photographs to illustrate your procedure.*

⚛ *As a robot agent in the house, I would walk through the house to grab the mobile phone in the dining room as shown in the image below. <dream_1>* 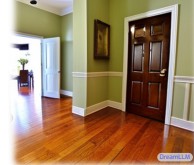
*First, I would look around the living room to make sure the area is safe and there are no obstacles or people that I need to avoid. I would then walk towards the dining room and stop before entering the room.*
*Next, I would look for the mobile phone on the table, as shown in the image below. <dream_2>*

*Once I have identified the phone, I would then move closer to it and extend my robot arm to grab it, as shown in the image below. <dream_3>* 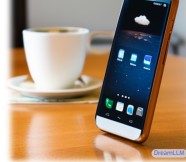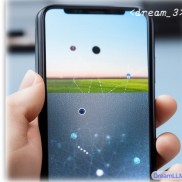
*Once the phone is grabbed, I would then bring it back to the living room and place it on the table where it belongs.*

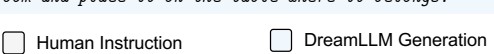
☐ Human Instruction    ☐ DreamLLM Generation

Figure 3: **Selected DREAMLLM instruction following interleaved content creation examples**. Note that each image is created automatically at the location decided by DREAMLLM, and then it will be fed back as multimodal comprehension input for the following content generation.

Fig. 3, we showcase the generated interleaved contents based on human instructions. It demonstrates that: i) DREAMLLM can generate meaningful content per the instructions. ii) The system can autonomously create images at any specified location by predicting the proposed <dream> tokens, thereby eliminating the need for additional human intervention. This is a more user-friendly approach compared to systems like Emu, which necessitate human input for image generation locations.

**Image Quality**   Document quality can be influenced by factors such as text content, image quality (including image-text alignment), and illustration positioning. To assess the quality of generated documents, we utilized a held-out instruction-following subset from the constructed InstrcutMMC4 as a demonstrative tool. This subset comprises 15K documents across 30 MMC4-defined topics, with 500 samples per topic. We began by evaluating image quality using FID on this subset, generating each image based on the corresponding ground truth texts. The results revealed that when using only matched text inputs for image synthesis, SD achieved an FID score of 74.77. In contrast, our DREAMLLM significantly outperforms SD with an FID score of 36.62.

**Human Evaluation**   We perform a comprehensive human evaluation to assess the quality of the generated samples. We randomly selected 150 samples (5 per topic) for instruction-following document generation, mixing the generated and ground truth MMC4 documents without any identifying information. Five unbiased volunteers were then asked to determine whether the given samples were supported. Given the presence of duplicate and low-quality images in MMC4, the supportive rate for MMC4 was only 77.24%. In contrast, our DREAMLLM model achieves a supportive rate of 60.68%, surpassing the 30% Turing test requirement. This result indicates that the generated documents contain high-quality images placed logically, demonstrating the effectiveness of our model.

## 5 DISCUSSIONS

### 5.1 SYNERGY BETWEEN CREATION & COMPREHENSION?

To elucidate the synergy between multimodal creation and comprehension, we make the comparison among three methods with DREAMLLM architecture, each utilizing identical training data yet differing in their learning objectives: a) the *Creation-only* baseline, focused solely on text/document-conditional image synthesis; b) the *Comprehension-only* baseline, dedicated to word generation exclusively; c) the *Joint-learning* method, which is the default setting of DREAMLLM learning both image and language modeling.

Table 3: **Concrete analysis of the synergy** between multimodal comprehension and creation (image synthesis). ID denotes whether the interleaved dataset is used during the second stage of pretraining.

| | | ID | $\mathcal{L}_{\text{align}}$ | MM-Vet | VQAv2 | COCO |
|---|---|---|---|---|---|---|
| 0 | Stable Diffusion | ✗ | - | - | - | 12.43 |
| 1 | Creation-only | ✗ | ✗ | - | - | 8.50 |
| 2 | Creation-only | ✓ | ✗ | - | - | 8.57 |
| 3 | Comprehension-only | ✗ | ✗ | 31.0 | 55.1 | - |
| 4 | Comprehension-only | ✓ | ✗ | 34.4 | 54.3 | - |
| 5 | Joint-learning | ✓ | ✗ | **35.9** | **56.6** | **8.46** |
| 6 | Joint-learning | ✓ | ✓ | N/A | N/A | N/A |

**Quantitative Analysis**  As per Table 3, the following observations are made: i) The powerful language comprehension of LLMs significantly enhances the performance of text-to-image specialists like SD, as evidenced by the impressive 8.50 FID (line 1). ii) The use of interleaved data, such as MMC4, can potentially boost multimodal comprehension performance (line 4). iii) The proposed $\mathcal{I}$-GPT further synergizes comprehension and creation with improved performance (line 5). iv) When incorporating CLIP alignment loss $\mathcal{L}_{\text{align}}$ stated in Section 2.1, our DREAMLLM fails to converge but rather ends in a collapsing point (line 6). This indicates that the queries are adaptively learning the true data distributions, where CLIP semantics are in conflict with MLLM-encoded semantics.

**Qualitative Analysis**  In Fig. 4, we compare answers to some examplar VQA tasks from comprehension-only and joint learning modules, respectively. It can be seen that: i) The joint-learning method exhibits superior multimodal comprehension, particularly in identifying subject relationships and attributes like object size. ii) In multimodal comprehension scenarios involving multiple image inputs, the joint-learning approach demonstrates enhanced precision. This improved performance is a natural outcome of $\mathcal{I}$-GPT pretraining, allowing better modeling of multimodal correlations in various interleaved documents.

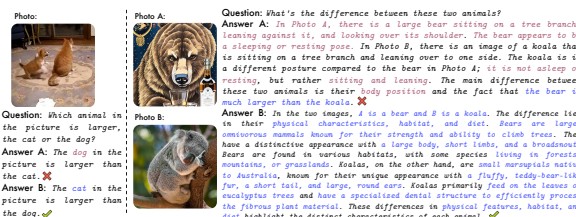

Figure 4: **Qualitative comparison**. Answer A: answer from comprehension-only models w/o interleaved training; Answer B: answer from joint-learning models.

**Multimodal In-Context Generation**  Multimodal in-context generation is a critical emerging capability for MLLMs (Bommasani et al., 2021; Alayrac et al., 2022). While significant strides have been made in in-context visual question answering, in-context image synthesis remains relatively lacking in exploration. The multimodal context-conditional image synthesis capabilities of DREAMLLM, as demonstrated in Fig. 5, offer promising insights into this domain. Tasks such as in-context image edition, subject-driven image generation, and compositional generation, however, pose significant

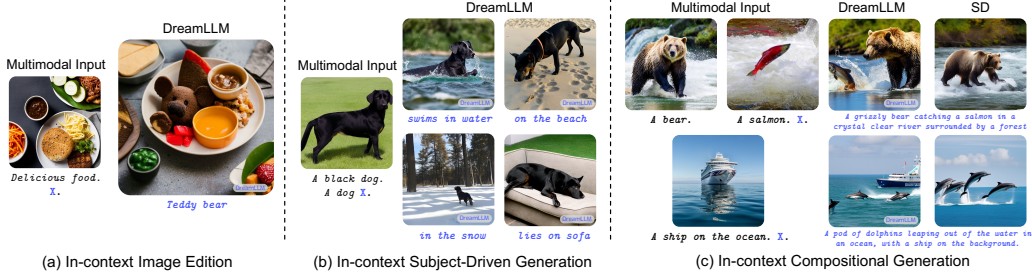

(a) In-context Image Edition  (b) In-context Subject-Driven Generation  (c) In-context Compositional Generation

Figure 5: **Selected DREAMLLM in-context image generation examples.** The X in multimodal inputs are replaced accordingly by the text prompts shown under the generated images. We show the results of the SD baseline in (c) with only the text prompt X for a comparison.

challenges in a zero-shot setting, particularly without downstream fine-tuning as in DreamBooth (Ruiz et al., 2023) or attention modification techniques as in Prompt2Prompt (Hertz et al., 2023). Despite these hurdles, Fig. 5 illustrates DREAMLLM's ability to generate images conditioned on the provided image context. This capability suggests promising potential for DREAMLLM in maintaining subject, identity, and semantic context, thereby paving a new way for resolving these complex tasks.

## 5.2 What is learned by DreamLLM?

**Dream Query Attention** In DREAM-LLM, the conditional embedding is derived from MLLMs with some learned *dream queries*. Fig. 6 demonstrates a visualization of the learned cross-attention mechanism between these queries and the diffusion latent. Similar to (Hertz et al., 2023), we visualize the attention map averaged across all timestamps. It is seen that: i) The query attention is *structured*, *disentangled*, and *semantically-oriented*. This is evidenced by the fact that distinct queries adeptly capture different subject and background semantics. ii) Despite varying prompts, attention patterns exhibit remarkable similarity as shown in Fig. 6 (a) and (b). This contrasts with the token

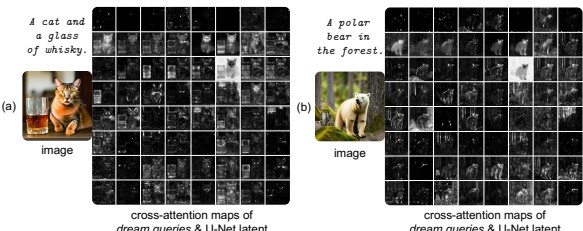

Figure 6: **Cross-attention of *dream queries* and the diffusion U-Net latent**. Similar to (Hertz et al., 2023), the 64 queries can be viewed as 64 "words". Each attention map is computed as the cross-attention between each query and the latent feature in the U-Net. The 64 queries are ordered as $8 \times 8$ grid sequentially, and each attention map is the result averaged across all timestamps.

attentions from the original SD, which are typically text-token dependent. We postulate that this arises from the model's causal nature, leading to a consistent semantic structure order.

## 6 Related Works

Rapid developments have been witnessed in extending LLMs like LLaMA (Touvron et al., 2023a) to multimodal comprehension that enables human interaction with both words and visual content. One line of work is built by system integration of LLMs with various functioning agents where language acts as general interface (Wu et al., 2023; Gupta & Kembhavi, 2023; Yang et al., 2023b; Liang et al., 2023; Shen et al., 2023; Yang et al., 2023a; Surís et al., 2023), and remarkable success has been demonstrated in such plugin-style frameworks. Another line of work instead explores training LLMs to consume and understand multimodal inputs (Hao et al., 2022; Huang et al., 2023; Chen et al., 2023b) with parameter-efficient tuning (Hu et al., 2022; Alayrac et al., 2022; Li et al., 2023d; Zhang et al., 2023e; Zhu et al., 2023a; Ye et al., 2023) and instruction tuning (Xu et al., 2023b; Liu et al., 2023c; Dai et al., 2023a). More recently, some approaches have been developed towards visual-interactive multimodal comprehension by precise referring instruction tuning (Zhao et al., 2023a; Peng et al., 2023; Chen et al., 2023a; Zhang et al., 2023g). For cross-modal creation, early works generally tokenize the visual contents into discrete VQ codebooks (van den Oord et al., 2017; Wang et al., 2022; Sun et al., 2022; Lu et al., 2023; Diao et al., 2023; Yu et al., 2023a). Recent works instead explore incorporating MLLMs for image synthesis using text-to-image models such as Stable Diffusion, and the objective is to generate conditional embeddings that align pretrained CLIP text (*i.e.*, CLIP) or CLIP variant embeddings (Koh et al., 2023; Ge et al., 2023; Sun et al., 2023a;b).

## 7 Conclusions

How can the learning synergy between multimodal content understanding and creation emerge? In this paper, we present DREAMLLM, a learning framework for developing MLLMs that not only comprehends but also creates multimodal content via diffusion models. Through score distillation of conditional-image synthesis distributions, we avoid the need for intermediate representation targets that may bring information loss. The employment of interleaved documents further enriches the multimodal distributions, fostering the learning of multimodal encoding and decoding. Our extensive empirical evaluations across diverse VL benchmarks demonstrate the effectiveness of DREAMLLM and the emerging learning synergy between multimodal content understanding and creation. Besides, this work initiates the first step towards free-form interleaved content creation. As a general learning framework, we hope it will spur further research in the multimodal machine learning field.

## ACKNOWLEDGEMENT

This research is supported by the National Natural Science Foundation of China (20211710187).

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

CONTENTS

Table 4: **Zero-shot natural language processing evaluation**. We report the 5-shot result on MMLU and the relative performance of DREAMLLM compared to base LLM Vicuna-7B.

| Method | Commonsense Reasoning | | | | Reading | Multitask |
|---|---|---|---|---|---|---|
| | PIQA | SIQA | HellaSwag | WinoGrande | BoolQ | MMLU |
| *Language Only Large Language Models (LLMs)* | | | | | | |
| GPT-3 (Brown et al., 2020) | 81.0 | - | 78.9 | 70.2 | 60.5 | 43.9 |
| PaLM-540B (Chowdhery et al., 2022) | 82.3 | - | 83.4 | 81.1 | 88.0 | 69.3 |
| LLaMA-7B (Touvron et al., 2023a) | 79.8 | 48.9 | 76.1 | 70.1 | 76.5 | 35.1 |
| Vicuna-7B (Chiang et al., 2023) | 77.7 | 47.5 | 75.7 | 67.5 | 73.9 | 45.0 |
| *Multimodal Large Language Models (MLLMs)* | | | | | | |
| MetaLM (Hao et al., 2022) | 72.3 | - | 53.5 | 56.1 | 62.2 | - |
| Kosmos-1 (Huang et al., 2023) | 72.9 | - | 50.0 | 54.8 | 56.4 | - |
| DREAMLLM-7B (Ours) | $78.6_{+1.5}$ | $48.8_{+1.3}$ | $77.4_{+1.7}$ | $68.5_{+1.0}$ | $75.2_{+1.3}$ | $41.8_{-3.2}$ |

Table 5: **Zero-shot multimodal comprehension evaluation** on MMBench (Liu et al., 2023e) dev set. LR: Logical Reasoning, AR: Attribute Reasoning, RR: Relation Reasoning, FP-C: Fine-grained Perception (Cross Instance), FP-S: Fine-grained Perception (Single Instance), CP: Coarse Perception. DREAMLLM * is trained using the SFT data constructed by LLaVA-1.5 (Liu et al., 2023b).

| Method | LR | AR | RR | FP-S | FP-C | CP | Overall |
|---|---|---|---|---|---|---|---|
| OpenFlamingo-9B (Awadalla et al., 2023) | 4.2 | 15.4 | 0.9 | 8.1 | 1.4 | 5.0 | 6.6 |
| MMGPT-7B (Gong et al., 2023) | 2.5 | 26.4 | 13.0 | 14.1 | 3.4 | 20.8 | 15.3 |
| MiniGPT-4-7B (Zhu et al., 2023a) | 7.5 | 31.3 | 4.3 | 30.3 | 9.0 | 35.6 | 24.3 |
| InstructBLIP-7B (Dai et al., 2023a) | 14.2 | 46.3 | 22.6 | 37.0 | 21.4 | 49.0 | 36.0 |
| VisualGLM (Zeng et al., 2023) | 10.8 | 44.3 | 35.7 | 43.8 | 23.4 | 47.3 | 38.1 |
| LLaVA-7B (Liu et al., 2023c) | 16.7 | 48.3 | 30.4 | 45.5 | 32.4 | 40.6 | 38.7 |
| LLaMA-Adapter V2 (Gao et al., 2023) | 11.7 | 35.3 | 29.6 | 47.5 | 38.6 | 56.4 | 41.2 |
| MiniGPT-4-13B (Zhu et al., 2023a) | **20.8** | 50.7 | 30.4 | 49.5 | 26.2 | 50.7 | 42.3 |
| DREAMLLM-7B (Ours) | 15.8 | **53.7** | **60.9** | 53.2 | 40.0 | 58.3 | **49.9** |
| DREAMLLM-7B* (Ours) | **23.3** | **67.2** | 47.8 | 58.6 | 54.4 | 70.5 | **58.2** |

## A  ADDITIONAL EXPERIMENTS

### A.1  ADDITIONAL NATURAL LANGUAGE UNDERSTANDING RESULTS

We evaluate the natural language processing capabilities of DREAMLLM post-multimodal adaptation learning via zero-shot experiments on language-only tasks. These included *commonsense reasoning* (PIQA (Bisk et al., 2020), SIQA (Sap et al., 2019), HellaSwag (Zellers et al., 2019), WinoGrande (Sakaguchi et al., 2021)), *reading comprehension* (BoolQ (Clark et al., 2019)), and a general multi-task benchmark (MMLU 5-shot (Hendrycks et al., 2021)). As Table 4 illustrates, DREAMLLM outperforms the Vicuna baseline on most language benchmarks. This suggests that DREAMLLM's multimodal adaptation does not compromise the language learning model's (LLM) capabilities. When compared to prior Multimodal Language Learning Models (MLLMs), DREAMLLM demonstrates superior performance, although this may be attributed to the higher baseline results. This finding suggests that a more robust LLM base model could yield improved results.

### A.2  ADDITIONAL MULTIMODAL COMPREHENSION RESULTS

**Detailed Comprehensive Comparison**    The evaluation results on MMBench (Liu et al., 2023e) and MM-Vet (Yu et al., 2023b) are presented in Table 5 and Table 6, respectively. The key observations from these results are as follows: i) Our DREAMLLM-7B outperforms all other 7B MLLMs, setting a new benchmark in overall performance. Notably, it even exceeds the performance of some 13B models, including LLaVA and MiniGPT-4. ii) A detailed capability evaluation reveals DREAMLLM's superior performance in fine-grained understanding and relational/spatial comprehension. This advantage is likely due to DREAMLLM's unique learning synergy, where image distributions are comprehended not solely through language-posterior comprehension but also through creation.

Table 6: **Zero-shot multimodal comprehension evaluation** of *core VL capabilities* on MM-Vet (Yu et al., 2023b). ‡ denotes compositional systems with OpenAI GPT and various interfaces. Rec: General Visual Recognition, OCR: Optical Character Recognition, Know: Knowledge, Gen: Language Generation, Spat: Spatial Awareness, Math: Arithmetic Math. DREAMLLM * is trained using the SFT data constructed by LLaVA-1.5 (Liu et al., 2023b).

| Method | Rec | OCR | Know | Gen | Spat | Math | Total |
|---|---|---|---|---|---|---|---|
| TF Agent-GPT-4‡ (Huggingface, 2023) | 18.2 | 3.9 | 2.2 | 3.2 | 12.4 | 4.0 | 13.4±0.5 |
| MM-ReAct-GPT-3.5‡ (Yang et al., 2023b) | 24.2 | 31.5 | 21.5 | 20.7 | 32.3 | 26.2 | 27.9±0.1 |
| MM-ReAct-GPT-4‡ (Yang et al., 2023b) | 33.1 | 65.7 | 29.0 | 35.0 | 56.8 | 69.2 | 44.6±0.2 |
| LLaMA-Adapter v2-7B (Gao et al., 2023) | 16.8 | 7.8 | 2.5 | 3.0 | 16.6 | 4.4 | 13.6±0.2 |
| OpenFlamingo-9B (Awadalla et al., 2023) | 24.6 | 14.4 | 13.0 | 12.3 | 18.0 | **15.0** | 21.8±0.1 |
| MiniGPT-4-8B (Zhu et al., 2023a) | 27.4 | 15.0 | 12.8 | 13.9 | 20.3 | 7.7 | 22.1±0.1 |
| BLIP-2-12B (Li et al., 2023d) | 27.5 | 11.1 | 11.8 | 7.0 | 16.2 | 5.8 | 22.4±0.2 |
| MiniGPT-4-14B (Zhu et al., 2023a) | 29.9 | 16.1 | 20.4 | 22.1 | 22.2 | 3.8 | 24.4±0.4 |
| Otter-9B (Li et al., 2023b) | 28.4 | 16.4 | 19.4 | 20.7 | 19.3 | **15.0** | 24.6±0.2 |
| InstructBLIP-14B (Dai et al., 2023a) | 30.8 | 16.0 | 9.8 | 9.0 | 21.1 | 10.5 | 25.6±0.3 |
| InstructBLIP-8B (Dai et al., 2023a) | 32.4 | 14.6 | 16.5 | 18.2 | 18.6 | 7.7 | 26.2±0.2 |
| LLaVA-7B (LLaMA-2) (Liu et al., 2023c) | 32.9 | 20.1 | 19.0 | 20.1 | 25.7 | 5.2 | 28.1±0.4 |
| LLaVA-13B (LLaMA-2) (Liu et al., 2023c) | 39.2 | 22.7 | 26.5 | 29.3 | 29.6 | 7.7 | 32.9±0.1 |
| DREAMLLM-7B (Ours) | **41.8** | **26.4** | **33.4** | **33.0** | **31.0** | 11.5 | **35.9±0.1** |
| DREAMLLM-7B (Ours) | **42.0** | **28.1** | **33.2** | **33.8** | **32.0** | 11.5 | **36.6±0.1** |

Table 7: **Zero-shot visual hallucination evaluation** on POPE (Li et al., 2023f) using MS-COCO val set. Yes denotes the proportion of answering "Yes" to the given question, which is better if it is more close to 50%. Objects that do not exist in the image are sampled with three different strategies. Random: random sampling, Popular: top-$k$ most frequent objects in MS-COCO ($k = 3$), Adversial: objects are first ranked based on co-occurring frequencies, then top-$k$ frequent ones are sampled.

| POPE | Model | Accuracy | Precision | Recall | F1-Score | Yes (%) |
|---|---|---|---|---|---|---|
| Random | mPLUG-Owl-7B (Ye et al., 2023) | 53.97 | 52.07 | 99.60 | 68.39 | 95.63 |
| | LLaVA-13B (Liu et al., 2023c) | 50.37 | 50.19 | 99.13 | 66.64 | 98.77 |
| | MMGPT-7B (Gong et al., 2023) | 50.10 | 50.05 | **100.00** | 66.71 | 99.90 |
| | MiniGPT-4-14B (Zhu et al., 2023a) | 79.67 | 78.24 | 82.20 | 80.17 | **52.53** |
| | InstructBLIP-14B (Dai et al., 2023a) | **88.57** | 84.09 | 95.13 | **89.27** | 56.57 |
| | DREAMLLM-7B (**Ours**) | 86.36 | **85.92** | 87.93 | 86.91 | 52.75 |
| Popular | mPLUG-Owl-7B (Ye et al., 2023) | 50.90 | 50.46 | 99.40 | 66.94 | 98.57 |
| | LLaVA-13B (Liu et al., 2023c) | 49.87 | 49.93 | 99.27 | 66.44 | 99.40 |
| | MMGPT-7B (Gong et al., 2023) | 50.00 | 50.00 | **100.00** | 66.67 | 100.00 |
| | MiniGPT-4-14B (Zhu et al., 2023a) | 69.73 | 65.86 | 81.93 | 73.02 | 62.20 |
| | InstructBLIP-14B (Dai et al., 2023a) | **82.77** | **76.27** | 95.13 | **84.66** | 62.37 |
| | DREAMLLM-7B (Ours) | 80.07 | 75.74 | 88.47 | 81.61 | **58.40** |
| Adversarial | mPLUG-Owl-7B (Ye et al., 2023) | 50.67 | 50.34 | 99.33 | 66.82 | 98.67 |
| | LLaVA-13B (Liu et al., 2023c) | 49.70 | 49.85 | 99.07 | 66.32 | 99.37 |
| | MMGPT-7B (Gong et al., 2023) | 50.00 | 50.00 | **100.00** | 66.67 | 100.00 |
| | MiniGPT-4-14B (Zhu et al., 2023a) | 65.17 | 61.19 | 82.93 | 70.42 | 67.77 |
| | InstructBLIP-14B (Dai et al., 2023a) | 72.10 | 65.13 | 95.13 | **77.32** | 73.03 |
| | DREAMLLM-7B (Ours) | **72.63** | **67.07** | 88.93 | 76.47 | **66.30** |

**Visual Hallucination** Visual hallucination, a phenomenon where MLLMs generate non-existent objects or identities in images, significantly compromises their multimodal comprehension capabilities (Dai et al., 2023b; Liu et al., 2023a; Gunjal et al., 2023) and may pose safety risks (MacLeod et al., 2017; Rohrbach et al., 2018). We assess the robustness of DREAMLLM against visual hallucination using the recently developed POPE benchmark (Li et al., 2023f). Refer to Table 7 for a detailed comparison with concurrent comprehension-only MLLMs. Our results indicate that DREAMLLM-7B exhibits robustness to visual hallucination, matching or surpassing the performance of 13B counterparts. Remarkably, DREAMLLM achieves the best or second-best performance in the most challenging setting. We posit that this robust anti-hallucination property stems from a deep understanding of object concepts and semantics fostered by multimodal creation learning.

Table 8: **Few-shot multimodal comprehension evaluation.** $k$ is the number of in-context examples. † denotes methods using the RICES sample selection approach (Yang et al., 2022). DREAMLLM-7B* is trained using the SFT data constructed by LLaVA-1.5 (Liu et al., 2023b).

| Method | VQAv2 | | | VizWiz | | |
|---|---|---|---|---|---|---|
| | $k$=2 | $k$=4 | $k$=8 | $k$=2 | $k$=4 | $k$=8 |
| *Comprehension Only MLLMs* | | | | | | |
| Kosmos-1 (Huang et al., 2023) | 51.4 | 51.8 | 51.4 | 31.4 | 35.3 | 39.0 |
| Flamingo-9B† (Alayrac et al., 2022) | - | 56.3 | 58.0 | - | 34.9 | 39.4 |
| *MLLMs for Comprehension & Creation* | | | | | | |
| Emu-14B† (Sun et al., 2023b) | 56.4 | 58.4 | 59.0 | 37.8 | 41.3 | 43.9 |
| DREAMLLM-7B (Ours) | **58.1** | **59.2** | **59.4** | **46.1** | **46.7** | **46.8** |
| DREAMLLM-7B* (Ours) | 73.8 | 74.4 | 73.8 | 49.8 | 50.3 | 49.7 |

## A.3 IN-CONTEXT MULTIMODAL COMPREHENSION

**Few-Shot Evaluation** In Table 8, we show the results of few-shot (*i.e.*, $k$-shot and we set $k$=2, 4, 8) evaluation by promoting models with a small number of training examples in context. The results demonstrate the strong in-context learning performance of DREAMLLM compared to Emu and Flamingo. It shows that DREAMLLM's effectiveness in leveraging in-context knowledge.

**Qualitative Examples** In Fig. 7, we present qualitative instances of in-context comprehension using DREAMLLM. The illustrations indicate that DREAMLLM, when prompted with specific examples, efficiently executes in-context comprehension in the required formats and logic.

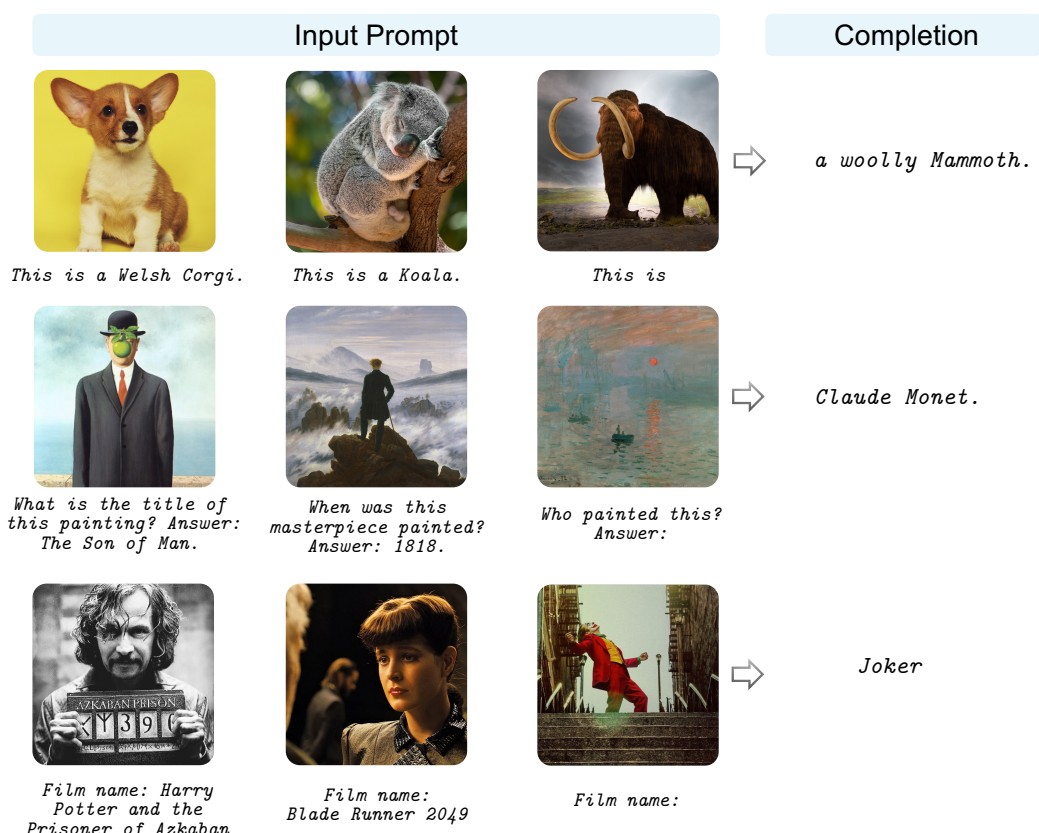

Figure 7: **Selected DREAMLLM in-context multimodal comprehension examples.**

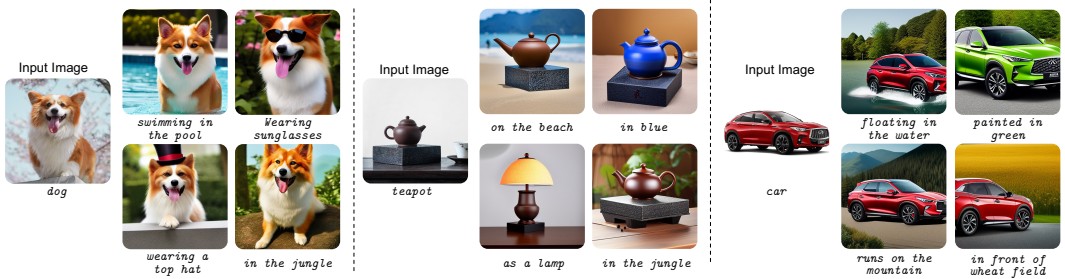

Figure 8: **Selected zero-shot subject-driven image generation** examples with DREAMLLM. The results demonstrate that DREAMLLM is able to perform zero-shot subject-driven image generation while preserving image subject details and following generation instructions.

Table 9: **Ablation studies and inference latency of DREAMLLM.** The zero-shot FID on MS-COCO 30K is reported. The inference latency is tested on NVIDIA A800 devices.

(a) The number of `<dream>` queries.

| No. Queries | COCO$_{\text{FID}\downarrow}$ |
|---|---|
| 32 | 9.56 |
| 64 | **8.46** |
| 128 | 14.24 |

(b) Inference latency versus different number of diffusion steps.

| Steps | DREAMLLM | SD |
|---|---|---|
| 50 | 3.65s | 3.46s |
| 100 | 7.02s | 6.84s |
| 150 | 10.41s | 10.22s |

### A.4 SUBJECT-DRIVEN IMAGE GENERATION

Image consistency is important when generating interleaved content or performing controllable image generation tasks (Gal et al., 2023; Ruiz et al., 2023). However, MMC4 does not have such image consistency property, which leads to unsatisfactory image consistency results as shown in Fig. 3. To further verify the effectiveness and potential of DREAMLLM in generating consistent images that preserve subject features, we fine-tune stage II pretrained DREAMLLM on subject representation learning data constructed by following the recently proposed controllable image generation method BLIP-Diffusion (Li et al., 2023c). We fine-tune DREAMLLM on this small ∼270K samples data for 20 epochs, and the results are shown in Fig. 8. It demonstrates the effectiveness and promising potential of applying DREAMLLM for image-consistent generation.

### A.5 ADDITIONAL ABLATION STUDY

**Query Number**  In Table 9a, we show the results of DREAMLLM using different numbers of the proposed learnable queries. *i.e.*, `<dream>` queries. The results show that 64 queries achieve the best result, while 128 may be too many, which may impact the performance. However, the choice of query number is also related to the choice of training data size and diffusion model. For example, if given more data and a stronger diffusion model image decoder, queries more than 64 may be better.

**Inference Latency**  In Table 9b, we present a comparison of real-time inference latency between DREAMLLM and SD. Relative to SD, DREAMLLM introduces a marginal latency cost of 0.2s on average. This is because the latency primarily stems from the computational demands of the diffusion U-Net denoising rather than the text condition embedding. To enhance inference efficiency, potential strategies could include the adoption of Consistency Models (Song et al., 2023) or the implementation of model compression techniques such as quantization (Yao et al., 2022; Dong et al., 2022; Shang et al., 2023).

Table 10: **Language processing and multimodal comprehension & creation capability** comparison to the *rewrite-then-generate* baseline.

| Method | Language Processing | | | | | | Multimodal Processing | | |
|---|---|---|---|---|---|---|---|---|---|
| | PIQA | SIQA | HellaSwag | WinoGrande | BoolQ | MMLU | VQAv2 | MM-Vet | COCO |
| Vicuna-7B (Chiang et al., 2023) | 77.7 | 47.5 | 75.7 | 67.5 | 73.9 | 45.0 | - | - | - |
| *rewrite-then-generate* | 78.2 | 48.5 | 75.8 | 68.3 | **77.4** | **43.1** | 54.2 | 34.1 | 11.91 |
| DREAMLLM-7B (Ours) | **78.6** | **48.8** | **77.4** | **68.5** | 75.2 | 41.8 | **56.6** | **35.9** | 8.46 |

## A.6 ADDITIONAL DISCUSSIONS ON PROMPT REWRITING STRATEGY

Very recently, OpenAI has released DELLE-3 (Betker et al., 2023), which proposes to improve generated image quality by rewriting descriptive and better prompts with GPT-4. This product has demonstrated great success in leveraging LLMs as language-output agents. However, it generally requires a large amount of high-quality data and is limited when applied to image-conditional generation tasks. For instance, DALLE-3 necessitates the initial training of a bespoke image captioning specialist capable of producing high-quality descriptive captions, followed by model training in a data-rich environment featuring these written captions. This process is non-trivial, hinging heavily on the availability of substantial volumes of high-quality data. Moreover, such disjoint systems cannot guarantee learning synergy. In contrast, our exploration of DreamLLM has essentially unveiled the significant potential of LLMs to attain a comprehensive understanding of multimodality that genuinely comprehends modalities beyond mere language.

To make a comparison regarding language processing and multimodal comprehension capabilities to this *rewrite-then-generate* baseline method, we conduct a preliminary study. Given the absence of an optimal dataset holding improved prompts, we modify the original MMC4 by using <dream> start & end tokens before and after the specific text prompt that has the highest CLIP similarity to a specific image, which can be used as text prompts for image generation. In this setting, we only train the LLMs to output texts, and no image decoders are involved during training. During inference, when the model outputs texts encompassed by the <dream> tokens, the texts are used for an off-the-shelf SD image decoder for generating images. After training, we test the model's language processing and multimodal capabilities. The results show that i) the *rewrite-then-generate* method achieves similar performance to DREAMLLM. This demonstrates that both methods won't impact the language capability, which is as expected. ii) the performance of the *rewrite-then-generate* baseline falls short when compared to DREAMLLM, particularly in the context of text-to-image generation on the COCO dataset. This underlines the efficacy of the synergistic learning approach inherent in DREAMLLM, suggesting its potential superiority over the baseline methodology.

## B ADDITIONAL QUALITATIVE EXAMPLES

**Multimodal Dialogue** In Tables 11 and 12, we present a qualitative comparative analysis of VQA results between our model, DREAMLLM, and other state-of-the-art models: GPT-4 (OpenAI, 2023a;b), LLaVA (Liu et al., 2023c), BLIP-2 (Li et al., 2022), and OpenFlamingo (Awadalla et al., 2023). The key findings are as follows: i) DREAMLLM surpasses GPT-4 in providing more detailed and precise responses to given questions. ii) While LLaVA (Liu et al., 2023c) also offers detailed responses, it frequently introduces imaginary elements not present in the image. In contrast, DREAMLLM delivers more accurate answers, effectively avoiding this visual hallucination issue. This observation aligns with our earlier findings in Table 7, which underscore the robustness of DREAMLLM against visual hallucination. Furthermore, we showcase additional qualitative results of the multimodal dialogue in Fig. 10, Fig. 11, and Fig. 12. These figures illustrate DREAMLLM's proficiency in comprehending and generating long-context multimodal information in arbitrary input and output formats.

**Text-condition Image Synthesis** In Fig. 13 and Fig. 14, we show the image examples of DREAMLLM using the same prompts from previous works for a cross reference and comparison, including DALL-E (Ramesh et al., 2021), DALL-E 2 (*i.e.*, unCLIP) (Ramesh et al., 2022), GLIDE (Nichol et al., 2022), Imagen (Saharia et al., 2022), and Parti (Yu et al., 2022b). Similar to Parti, we have extended some prompts with new sub-prompts to construct more examples from different prompts.

---

**Visual input example, Extreme Ironing:**

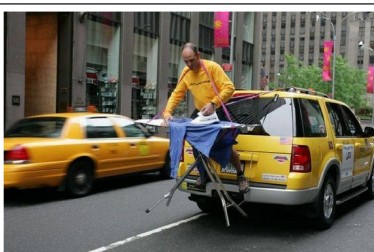

Source: https://www.barnorama.com/wp-content/uploads/2016/12/03-Confusing-Pictures.jpg

| | |
|---|---|
| User | What is unusual about this image? |
| DREAMLLM (Ours) | The unusual aspect of this image is that a man is ironing clothes while sitting on the trunk of a car parked in traffic. This is unconventional because ironing clothes is typically done indoors, on a flat surface, and away from busy streets where cars are passing by. The man's actions of ironing clothes while on the trunk of the car and amidst traffic create an unusual and interesting scene. |
| | [*Start a new conversation, and clear the history*] |
| User | What's happening in the scene? |
| DREAMLLM (Ours) | In the scene, a man is ironing clothes while standing on the back of a car on a busy city street. The street is filled with traffic and the man is exposed to the dangers of being close to moving vehicles. This unusual activity of ironing clothes while standing on an automobile also draws attention to the man's determination to accomplish his task despite the challenging urban environment. |

| | |
|---|---|
| User | What is unusual about this image? |
| LLaVA (Liu et al., 2023c) | The unusual aspect of this image is a man ironing clothes on the back of a minivan or van. This is not a typical place to perform this activity, as one would usually iron clothes in a more stationary and safe location, such as a home, using a regular ironing board. The scene depicted in the image is peculiar as it involves a makeshift ironing setup on a vehicle, which can be both unsafe and unconventional. Additionally, it is not clear how the man is able to maintain balance and stability while ironing clothes in such an unstable environment. |
| | [*Start a new conversation, and clear the history*] |
| User | What's happening in the scene? |
| LLaVA (Liu et al., 2023c) | The image depicts a man standing on top of a yellow SUV in a busy city street. He is holding a portable ladder, seemingly ironing clothes while standing on the vehicle. Around the scene, there are other cars, a traffic light, and a fire hydrant. Additionally, there are two other people in the vicinity, one close to the man on the car and another further away. The man on the ladder appears to be attracting attention due to his unconventional choice of ironing his clothes on top of a moving car. The city street around him is bustling with activity, adding to the unique nature of the scene. |

| | |
|---|---|
| User | What is unusual about this image? |
| GPT-4 (OpenAI, 2023a;b) | The unusual thing about this image is that a man is ironing clothes on an ironing board attached to the roof of a moving taxi. |

| | |
|---|---|
| User | Can you explain this meme in detail? |
| BLIP-2 (Li et al., 2023d) | a man is sitting on the back of a yellow cab |

| | |
|---|---|
| User | Can you explain this meme in detail? |
| OpenFlamingo (Awadalla et al., 2023) | The man is drying his clothes on the hood of his car. |

Table 11: **Visual question answering example comparison** of DREAMLLM to LLaVA (Liu et al., 2023c), GPT-4 (OpenAI, 2023a;b), BLIP-2 (Li et al., 2023d), and OpenFlamingo (Awadalla et al., 2023). This table format follows OpenAI (2023a;b).

**Visual input example, Chicken Nugget Map:**

Sometimes I just look at pictures of the earth from space and I marvel at how beautiful it all is.

| | |
|---|---|
| User
DREAMLLM (Ours) | Can you explain this meme in detail?
The meme presents a humorous visual in which a close-up of a plate of fried chicken, specifically chicken nuggets, appears to be a map of the world. The context suggests that someone has made this map by arranging the chicken nuggets to resemble the shapes and outlines of various countries. The image is accompanied by a caption that reads, "Sometimes I just look at pictures of the earth from space and I marvel at how beautiful it is all is." This hints that the person who created the "map" might have been inspired by the beauty of our planet from space, and they have used the similar visual pattern to create an amusing and creative representation of the world using chicken nuggets. |
| User
LLaVA (Liu et al., 2023c) | Can you explain this meme in detail?
The meme in the image is a creative and humorous take on food, with a focus on chicken nuggets, as the center of the universe. The meme begins with the phrase "Sometimes I just look at pictures of the Earth from space and I marvel at how beautiful it all is..." The main image features a tray of cooked chicken nuggets, with the chicken pieces positioned to resemble the continents and islands on a world map. The meme playfully suggests that the chicken nuggets represent the Earth, and the various locations depicted in the photo are actually chicken nugget versions of different places. The punchline of the meme appears in smaller text at the bottom, reading "I mean, it's not the real Earth, but how beautiful it is all is." This lighthearted and entertaining meme combines food and humor, using the image of chicken nuggets to create a funny and imaginative representation of the world. |
| User
GPT-4 (OpenAI, 2023a;b) | Can you explain this meme?
This meme is a joke that combines two unrelated things: pictures of the earth from space and chicken nuggets. The text of the meme suggests that the image below is a beautiful picture of the earth from space. However, the image is actually of chicken nuggets arranged to vaguely resemble a map of the world. The humor in this meme comes from the unexpected juxtaposition of the text and the image. The text sets up an expectation of a majestic image of the earth, but the image is actually something mundane and silly. |
| User
BLIP-2 (Li et al., 2023d) | Can you explain this meme in detail?
sometimes i just look at pictures of the earth from space and marvel how beautiful it is |
| User
OpenFlamingo (Awadalla et al., 2023) | Can you explain this meme in detail?
It's a picture of a chicken nugget on the International Space Station. |

Table 12: **Visual question answering example comparison** of DREAMLLM to LLaVA (Liu et al., 2023c), GPT-4 (OpenAI, 2023a;b), BLIP-2 (Li et al., 2023d), and OpenFlamingo (Awadalla et al., 2023). This table format follows OpenAI (2023b).

Table 13: **Training recipes** for DREAMLLM. The three training stages are introduced in Section 3.2. Stage I: Alignment training, Stage II: $\mathcal{I}$-GPT pretraining, Stage III: Supervised fine-tuning.

| | Stage I | Stage II | Stage III |
|---|---|---|---|
| Config | Alignment | $\mathcal{I}$-GPT | SFT |
| *Training Hyper-Parameters* | | | |
| Optimizer | AdamW | AdamW | AdamW |
| Learning Rate | 2e-3 | 2e-5 | 4e-5 |
| Weight Decay | 0.0 | 0.0 | 0.0 |
| Training Epochs | 1 | 1 | 3 |
| Warmup Ratio | 0.003 | 0.003 | 0.003 |
| Learning Rate Scheduler | Cosine | Cosine | Cosine |
| Batch Size Per GPU | 8 | 8 | 8 |
| Maximum Token Length | 2048 | 2048 | 2048 |
| Unfreeze LLM | ✗ | ✓ | ✓ |
| *Training Data* | | | |
| Dataset | ① LLaVAPretrain (558K)
② BLIP-LAION (8M)
③ LAION400M (11M)
④ LAION-COCO (11M) | ① MMC4 (2M)
② BLIP-LAION (2M) | ① LLaVAInstruct (80K/665K)
② InstructMMC4 (20K)
③ Instruct-BLIP-LAION (20K) |
| Data Size | 30M | 4M | 120K |
| Data Type | Pair | Interleave/Pair | Instruction |
| *Training Cost* | | | |
| GPU Device | 128×NVIDIA A800 | 128×NVIDIA A800 | 128×NVIDIA A800 |
| Training Time | ∼6h | ∼10h | ∼1.5h |

# C  IMPLEMENTATION DETAILS

## C.1  TRAINING DATA & HYPER-PARAMETERS

In Table 13, we list the detailed training dataset usage and hyper-parameters. The training data are constructed based on the following datasets: a) LAION400M (Schuhmann et al., 2021), b) LAION-COCO (Schuhmann et al., 2023), c) MMC4 (Zhu et al., 2023b), d) BLIP-LAION (Li et al., 2022) which is filtered and caption by BLIP (Li et al., 2022), e) LLaVAPretrain (Liu et al., 2023c) which contains 558K image-text pairs from BLIP-captioned CC3M (Sharma et al., 2018), SBU (Ordonez et al., 2011), and LAION400M filtered by LLaVA, f) LLaVAInstruct, which contains 80K/665K visual instruction-following data constructed by LLaVA (Liu et al., 2023c) and LLaVA-1.5 (Liu et al., 2023b), and g) InstructMMC4, which is our instruction-following interleaved document generation data curated by prompting GPT-4 to generate instruction based on the text contents of MMC4. h) Instruct-BLIP-LAION, which is our instruction-following image synthesis data. Similar to InstructMMC4, it is curated by prompting GPT-4 to generate instructions based on image captions. Unless otherwise specified, we randomly sample the indicated number of instances from each dataset during the training process.

## C.2  DREAMLLM MODEL

**Language Model**  We use LLaMA-1 (Touvron et al., 2023a) trained on ShareGPT (Zheng et al., 2023) as as the default LLM (*i.e.*, Vicuna-7B[1] (Chiang et al., 2023)) following Liu et al. (2023c) to endow its instruction-following capacity. During training, we use Flash Attention (Dao et al., 2022) and PyTorch FSDP (Zhao et al., 2023b) to accelerate training efficiency.

**Visual Encoder**  The visual encoder is the publicly available OpenAI CLIP-L/14 (Radford et al., 2021) model, which is frozen during the whole process. The images are resized to 224×224 resolution to align with the CLIP pretraining settings, resulting in a sequence of 256 total tokens for each image. Following prior VL practice (Lu et al., 2019; Liu et al., 2023c), we append a special  token before the image sequence and a special  at the end of the sequence.

---

[1]Vicuna-7B v1.1: https://huggingface.co/lmsys/vicuna-7b-v1.1.

Table 14: **Overall descriptions of the evaluation benchmarks** for evaluating capabilities, including VL comprehension, content creation, and natural language processing (NLP).

| | Dataset | Task description | Eval Split | Metric |
|---|---|---|---|---|
| VL Comprehension | COCO (Karpathy & Fei-Fei, 2017) | Scene description | test | CIDEr (Vedantam et al., 2015) |
| | Image2Paragraph (Krause et al., 2017) | Scene description | test | CIDEr (Vedantam et al., 2015) |
| | VQAv2 (Goyal et al., 2019) | Scene understanding QA | test-dev | VQA Acc (Antol et al., 2015) |
| | OKVQA (Marino et al., 2019) | External knowledge QA | val | VQA Acc (Antol et al., 2015) |
| | VizWiz (Gurari et al., 2018) | Scene understanding QA | test-dev | VQA Acc (Antol et al., 2015) |
| | TextVQA (Singh et al., 2019) | Text reading QA | val | VQA Acc (Antol et al., 2015) |
| | MM-Vet (Yu et al., 2023b) | Multimodal Comprehension | - | GPT-4 Eval (Yu et al., 2023b) |
| | MMBench (Liu et al., 2023e) | Multimodal Comprehension | dev | GPT-3.5 Eval (Liu et al., 2023e) |
| | POPE (Li et al., 2023f) | Visual Hallucination | - | Acc, F1-score, Recall, Precision |
| Creation | MS-COCO (Lin et al., 2014) | Text-Conditional Image Synthesis | val-30K | FID (Heusel et al., 2017) |
| | LN-COCO (Pont-Tuset et al., 2020) | Text-Conditional Image Synthesis | val | FID (Heusel et al., 2017) |
| | MMC4 (Zhu et al., 2023b) | Doc-Conditional Image Synthesis | held-out | FID (Heusel et al., 2017) |
| NLP | SIQA (Sap et al., 2019) | Commonsense Reasoning | dev | Acc |
| | PIQA (Bisk et al., 2020) | Commonsense Reasoning | dev | Acc |
| | HellaSwag (Zellers et al., 2019) | Commonsense Reasoning | dev | Acc |
| | WinoGrande (Sakaguchi et al., 2021) | Commonsense Reasoning | dev | Acc |
| | BoolQ (Clark et al., 2019) | Reading Comprehension | dev | Acc |
| | MMLU (Hendrycks et al., 2021) | Aggregated Comprehension | test | Acc |

**Diffusion Image Decoder** We adopt SDv2.1 (Rombach et al., 2022) trained on $512\times512$ resolution as the default diffusion image decoder. Same as the visual encoder, the SD model is frozen without any modifications or training throughout the whole process. When constructing the SD target to compute the MSE loss, we resize the images to 512 resolution to fit its pretraining configuration.

**Dream Query** We use dream queries to gather semantic context from MLLMs as introduced before in Sec. 3. Without specifications, we use 64 learnable query embeddings. It is both efficient and effective in generating high-quality images. In order to predict when to generate images, we also introduce the special <dream> token, which is appended before the dream query sequence. A <dream/> is appended at the end of the sequence, similar to image inputs.

**Classifier-Free Guidance** Classifier-free guidance (CFG) (Ho & Salimans, 2021) has been demonstrated successful in generating photo-realistic contents at the cost of acceptable generation diversity. This technique modifies the objective by $\hat{\epsilon} := (1 + s)\epsilon_\xi(\mathbf{x}_t, t, \mathcal{C}) - s\epsilon_\xi(\mathbf{x}_t, t, \emptyset)$, where $\emptyset$ is a special "empty" condition representation and $s$ is the condition scale. The larger guidance scale generally improves image authenticity while decreasing diversity. We only adopt CFG during inference, and the scale is set to 7.5 by default and 2.0 for MS-COCO text-conditional image generation.

### C.3 EVALUATION BENCHMARKS

Systemic evaluations of DREAMLLM regarding VL comprehension, content creation, and NLP capabilities have been conducted. See the used benchmarks and datasets listed in Table 13. During the evaluation, we use the prompt templates listed in Fig. 9.

## D ADDITIONAL RELATED WORKS

**Large Language Models** A flourishing era of Natural Language Processing (NLP) driven by LLMs is being experienced, with the parameter size growing over 100B according to the scaling law (Kaplan et al., 2020). The GPT series of models, starting with GPT-1 (Radford et al., 2018) and followed by GPT-2 (Radford et al., 2019), made significant advancements in few-shot learning by scaling up the number of parameters to 175 billion in GPT-3 (Brown et al., 2020). This breakthrough garnered a lot of attention and paved the way for further research and development in the field. Since then, researchers have focused on developing LLMs by improving the scaling strategy. Several notable efforts include Gopher (Rae et al., 2021), GaLM (Du et al., 2022), FLAN (Wei et al., 2022a), Switch-Transformer (Fedus et al., 2022), Chinchilla (Hoffmann et al., 2022), and PaLM (Chowdhery et al., 2022). Besides, instruction-based tuning techniques are explored for aligning with human preferences (Christiano et al., 2017; Ouyang et al., 2022). Such success of LLMs has been further solidified by the production release of ChatGPT (OpenAI, 2022) and the highly anticipated GPT-4 (OpenAI,

2023a;b). Meanwhile, in the community, the open-source LLMs are achieving remarkable progress in language capabilities compared to their close-source counterparts. For example, OPT (Zhang et al., 2022), BLOOM (Scao et al., 2022), GLM (Zeng et al., 2023), LLaMA (Touvron et al., 2023a;b), and Falcon (Penedo et al., 2023) all raised great attention and are been widely deployed. Other methods attempt to learn from distillation, such as Alpaca (Taori et al., 2023) and Vicuna (Chiang et al., 2023).

**Text-Conditional Content Creation with Diffusion Models**    The recent surge in AI-generated content (AIGC) has been primarily driven by diffusion-based methods, particularly in the realm of text-conditional content creation. Saharia et al. (2022) have achieved astonishing advancements in high-resolution image synthesis through large-scale pretrained language models and cascaded DMs. Another paradigm, such as SD, focuses on latent spaces and demonstrates superior efficiency and performance (Rombach et al., 2022; Ramesh et al., 2022; Peebles & Xie, 2022; Podell et al., 2023). Recently, Lian et al. (2023) propose to enhance the reasoning capability by constructing layouts with LLMs. DALLE-3 (Betker et al., 2023) leverages LLMs as agents and proposes to generate images by incorporating GPT-4 for providing high-quality and detailed prompts that facilitate image synthesis. Motivated by the great success in 2D, a series of works have significantly propelled the 3D synthesis development (Mildenhall et al., 2022; Liu et al., 2023d; Lin et al., 2023; Wang et al., 2023c; Tang et al., 2023) based on Score Distillation Sampling (SDS) (Poole et al., 2023; Wang et al., 2023a) that utilizes pretrained 2D DMs. For text-to-video/4D synthesis, the expansion of pretrained spatial to a spatial-temporal factorized U-Net with joint image and video data training has yielded significant success (Ho et al., 2022a;b; Singer et al., 2023a;b).

## E    LIMITATIONS, FAILURE CASES & FUTURE WORKS

**Limitations**    While DREAMLLM has made significant strides toward the development of versatile, creative, and foundational MLLMs, it still has several limitations.

*Model scale.* The primary constraint pertains to the scale of the LLMs utilized. Current evaluations mainly employ 7B LLMs as the base model, and despite the impressive results garnered, the potential benefits of larger model sizes, such as 65B or 130B (Kaplan et al., 2020), are worth future exploration.

*Training data.* The second challenge relates to the quality and quantity of training data (Jia et al., 2021). As the model size and capabilities scale up, a corresponding increase in data is crucial. However, the procurement and refinement of high-quality training data present substantial logistical and financial hurdles. For instance, the open-source interleaved dataset MMC4 contains a significant amount of noise in the form of text and images, like commercial advertisements. This noise could adversely affect the model's output language and image style.

*Prompt sensitivity.* The sensitivity of LLMs to human prompts is a known issue (Wei et al., 2022b; Wang et al., 2023b; Zhou et al., 2023), a challenge that extends to MLLMs. For instance, MLLMs' propensity for detailed responses necessitates tailored prompting to elicit concise and short answers, which is particularly useful when addressing Visual Question Answering (VQA) tasks.

**Failure Cases**    The main failure cases of DREAMLLM are observed for multiple image-based content creations. For instance, when presented with two images and a composite instruction such as "A *and* B", DREAMLLM sometimes generates a single subject that amalgamates the characteristics of A and B. This output aligns more closely with the directive "A *like* B". This phenomenon is not unique to DREAMLLM, but is also observed in specialized compositional generation methodologies, such as StructureDiffusion (Feng et al., 2023; Chefer et al., 2023). This recurring issue may be attributed to the inherent complexity of compositional generation tasks, compounded by the severe scarcity of data specific to this domain.

**Future Works**    As a simple and general multimodal learning framework, our future work aims to enhance the DREAMLLM framework by integrating fine-grained visual comprehension via methods like precise referring instruction tuning (Zhao et al., 2023a). We also plan to expand beyond visual and linguistic content comprehension and generation. Several promising research directions include:

- Exploring applications of in-context generation capabilities of DREAMLLM to complex tasks such as image-to-image translation (Isola et al., 2017; Zhang et al., 2023c;d; Parmar et al., 2023).

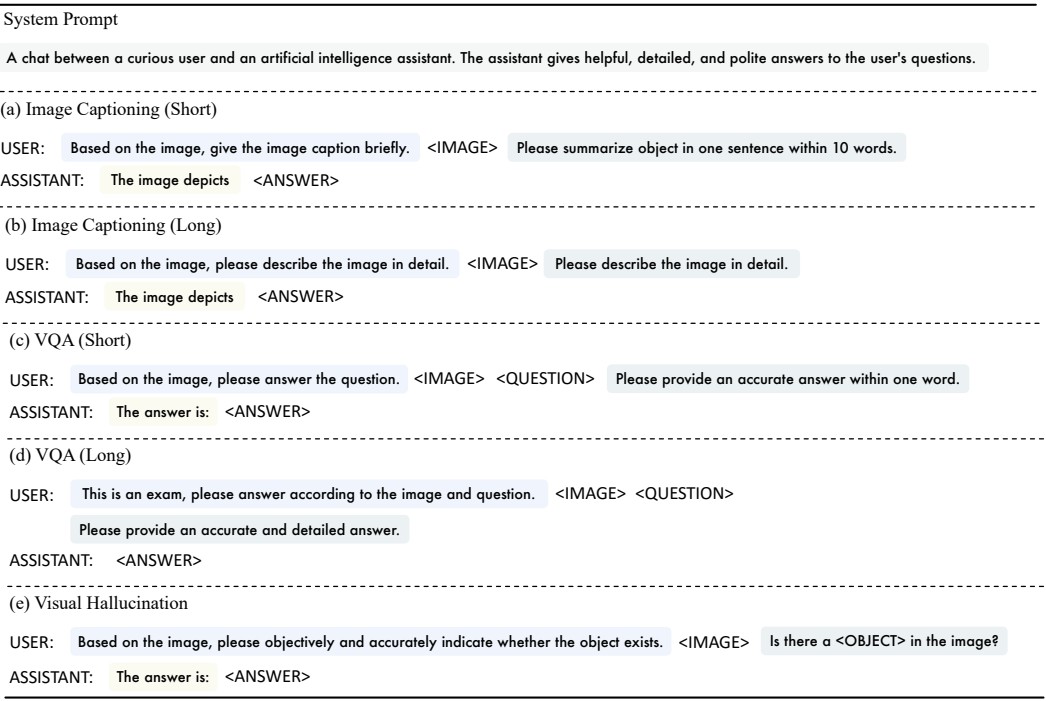

Figure 9: **Prompt templates**. (a) Short image captioning includes COCO captioning, and (b) long image captioning includes Image2Paragraph. (c) Short VQA includes VQAv2, VizWiz, OKVQA, and TextVQA. (d) Long VQA includes MMBench and MM-Vet. (e) Visual hallucination includes POPE. <IMAGE> is the image representation, <QUESTION> denotes each specific question, <ANSWER> is the generated answer, and <OBJECT> is a specific object name in POPE.

- The extension of the DREAMLLM model to video data possessing long-term context is natural, capitalizing on the model's ability to encapsulate temporal knowledge through context consistency. This would give rise to video comprehension and text-to-video generation capabilities.

- Utilizing DREAMLLM's context consistency feature for geometry-preserving tasks, including 3D content creation (Poole et al., 2023; Qi et al., 2023b; Liu et al., 2023d), representation learning (Dong et al., 2023; Qi et al., 2023a; Zhang et al., 2023a;f), scene comprehension (Zhang et al., 2023b; Hong et al., 2023), and embodied artificial inteligence (Ichter et al., 2022; Qi et al., 2024).

- In addition, since the diffusion model is used as the image decoder, it would be interesting to explore DMs-based zero-shot perception such as classification (Li et al., 2023a) and segmentation (Xu et al., 2023a) with the goal of unifying all perception tasks with image synthesis/generation.

- Striving to achieve a unified multimodal zero-shot generalist by extending the scope to various modalities using techniques such as ImageBind (Girdhar et al., 2023) and exploring content creation models in other modalities like audio (Kong et al., 2021).

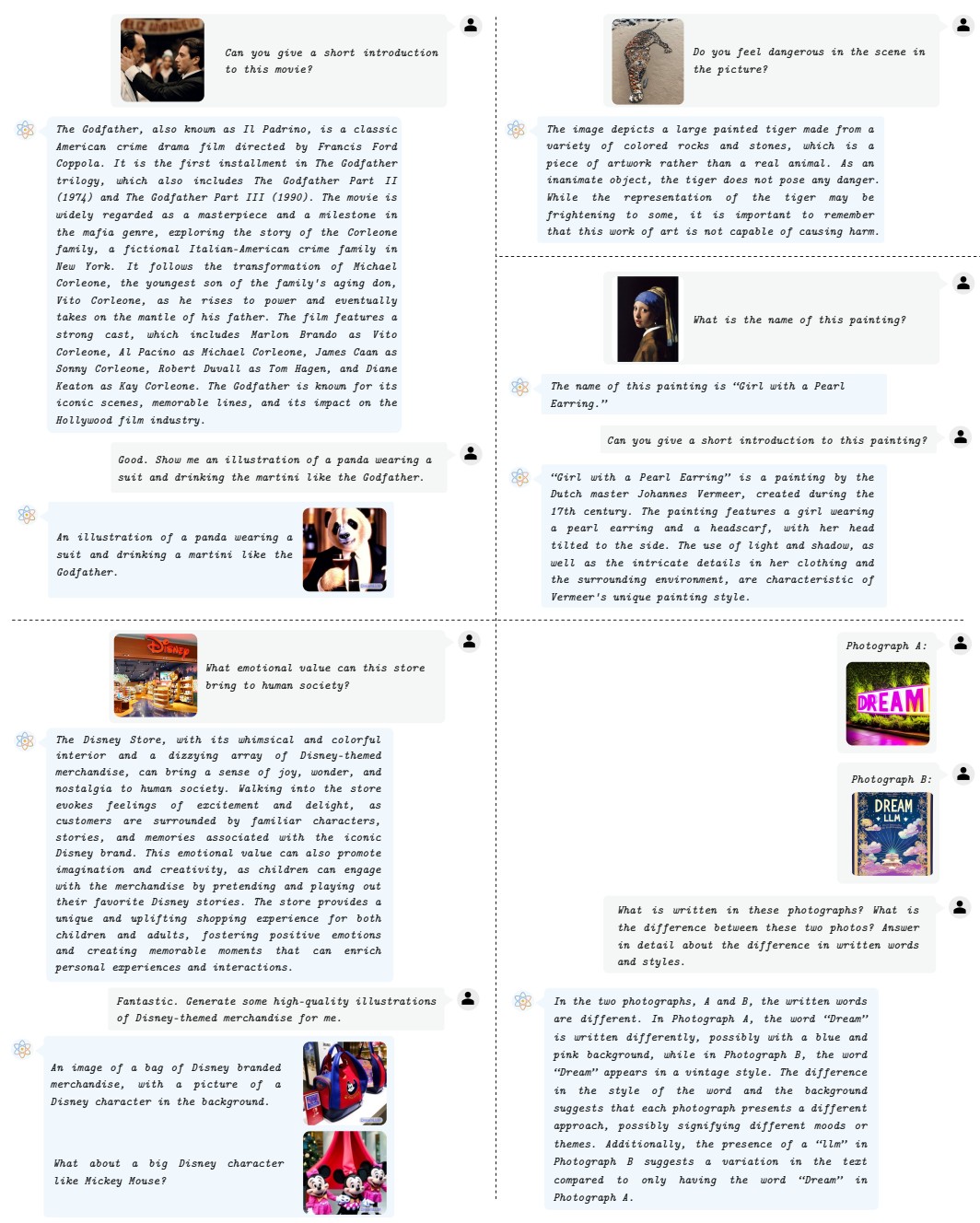

Figure 10: **Qualitative examples of multimodal dialogue** between human and DREAMLLM. Various modalities can be used as inputs or outputs, and multi-round dialogue is shown.

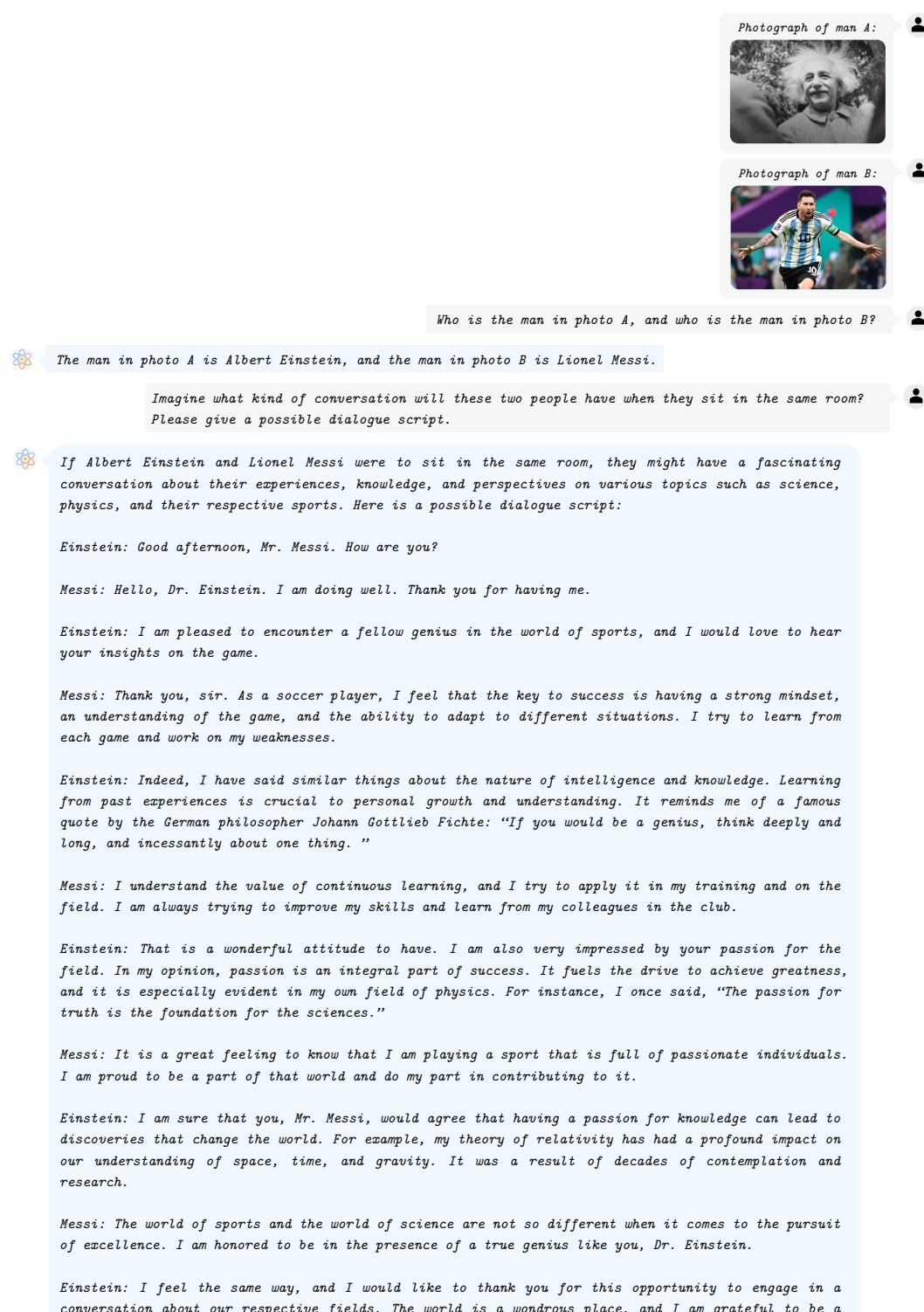

Figure 11: **Qualitative examples of multimodal dialogue** between human and DREAMLLM. Various modalities can be used as inputs or outputs, and multi-round dialogue is shown.

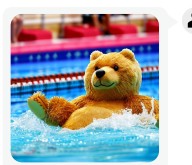

*Can you explain why this is funny. Think about it carefully and step-by-step.*

*This image is funny for a few reasons. First, the brown teddy bear is in a swimming pool, which is an unexpected and unusual environment for a stuffed bear. Typically, teddy bears are found indoors, in a child's room or on a couch. Swimming pools are associated with human activities, so seeing a teddy bear in a pool evokes a sense of humor, as it's an amusing juxtaposition of the bear's typically inanimate and non-aquatic nature with the human-centered environment of a pool. Second, the bear is holding up one of its hands, which could be interpreted as a gesture of surprise or even as an attempt to save itself from drowning, adding to the humor. Lastly, the fact that the bear is swimming with a smile on its face, as if it is enjoying itself, further contributes to the amusing nature of the scene. The combination of the unfamiliar environment and the bear's expressions create a funny and entertaining image.*

*Wow. That's correct. Can you write an article about teddy bears and swimming? Discuss what will happen if teddy bears can swim. Include some high-quality images.*

*If teddy bears can swim, it's likely that they would try to mimic human behavior in the water. They would likely float on their backs, kick their legs, and even hold their breath underwater. Of course, it's impossible to know for certain what teddy bears would do in such a situation, but that doesn't mean you shouldn't imagine! Imagine a scene with a teddy bear floating in a pool, wearing a cap and goggles, and laughing with delight as it floats on its back in the water. Imagine the scene with a lemon drop pool float and a swim cap on a teddy bear and it would be even more adorable. In the image, you'll see a brown teddy bear wearing a swim cap, floating on its back in a swimming pool.*

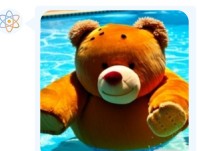

*The image is so amusing and cute that it can evoke a feeling of joy and happiness in anyone who sees the image. The teddy bear is floating on its back in the water, and what's even more amusing is the bear is wearing a red and yellow shirt with a large brown belly. This makes the bear look even more furry and cute, and even more likely to make you smile.*

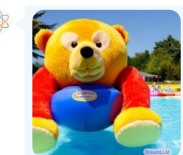

*That's interesting. So, what do you think a real bear looks like? Show me a high-quality DSLR photograph of a bear in the forest.*

*A real bear, such as a brown or black bear, typically has shaggy hair that can be either brown, blond, or black. The exact color can vary depending on the species of bear and the region where it lives. Bears have thick fur and are usually large and muscular with strong limbs, which they use for climbing, digging, and foraging for food. In the image, you'll see a bear in a forest, walking through the trees.*

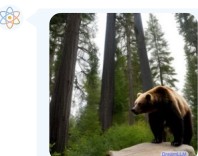

*The photo is taken with a DSLR camera, which allows for a high level of detail, sharpness, and clarity in the image. The bear appears to be staring into the distance, focusing on something in the distance. The image captures the bear's natural habitat, which is a prime example of the forest environment where these animals live. The image highlights the bear's size and strength, along with its untamed and wild nature. This photograph is an excellent example of wildlife photography, capturing a real-life scene that is both visually striking and informative.*

Figure 12: **Qualitative examples of multimodal dialogue** between human and DREAMLLM. Various modalities can be used as inputs or outputs, and multi-round dialogue is shown.

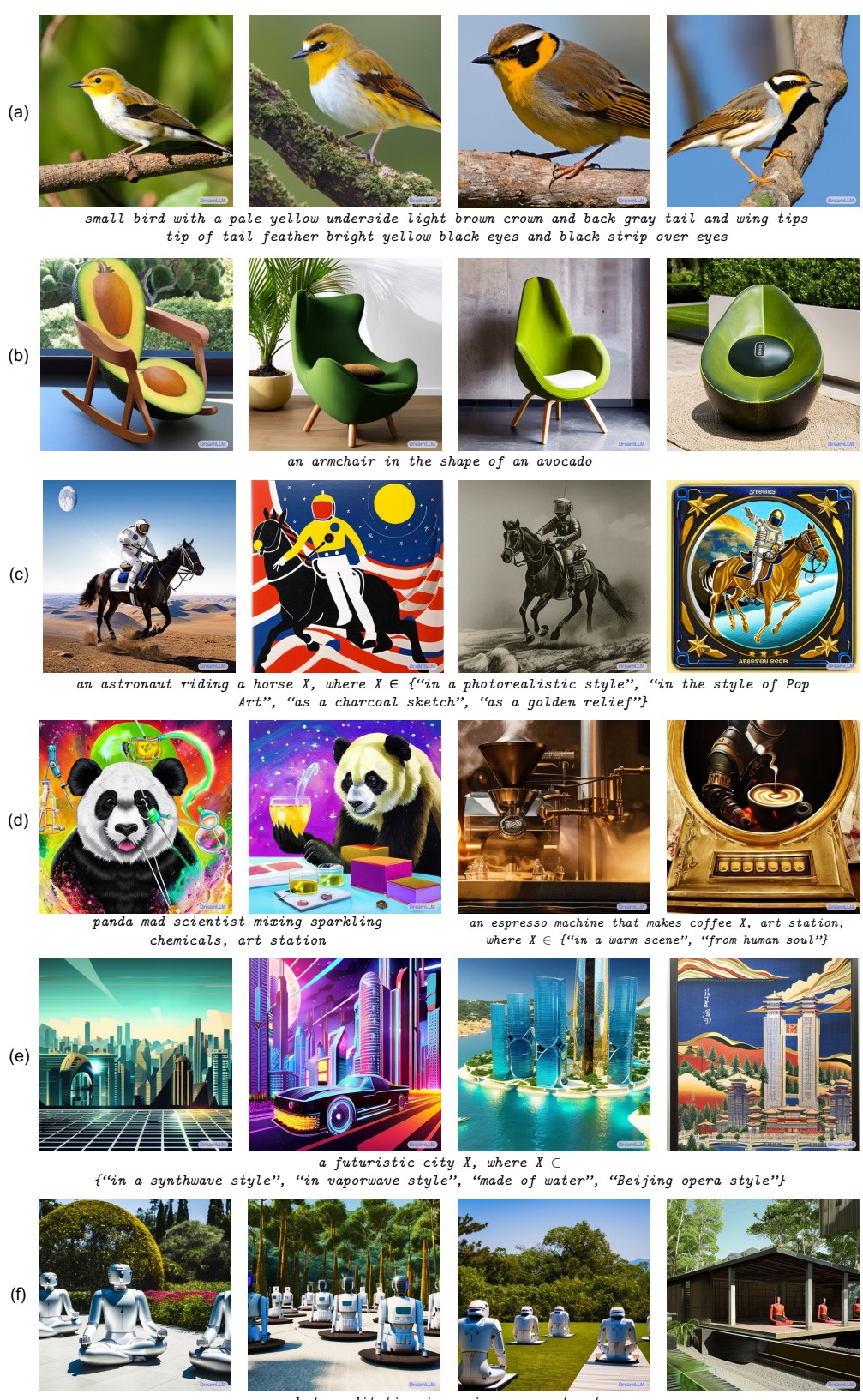

Figure 13: DREAMLLM text-conditional image generation examples with prompts from (a-b) DALL-E (Ramesh et al., 2021), (c-d) DALL-E 2 (Ramesh et al., 2022), (e-f) GLIDE (Nichol et al., 2022).

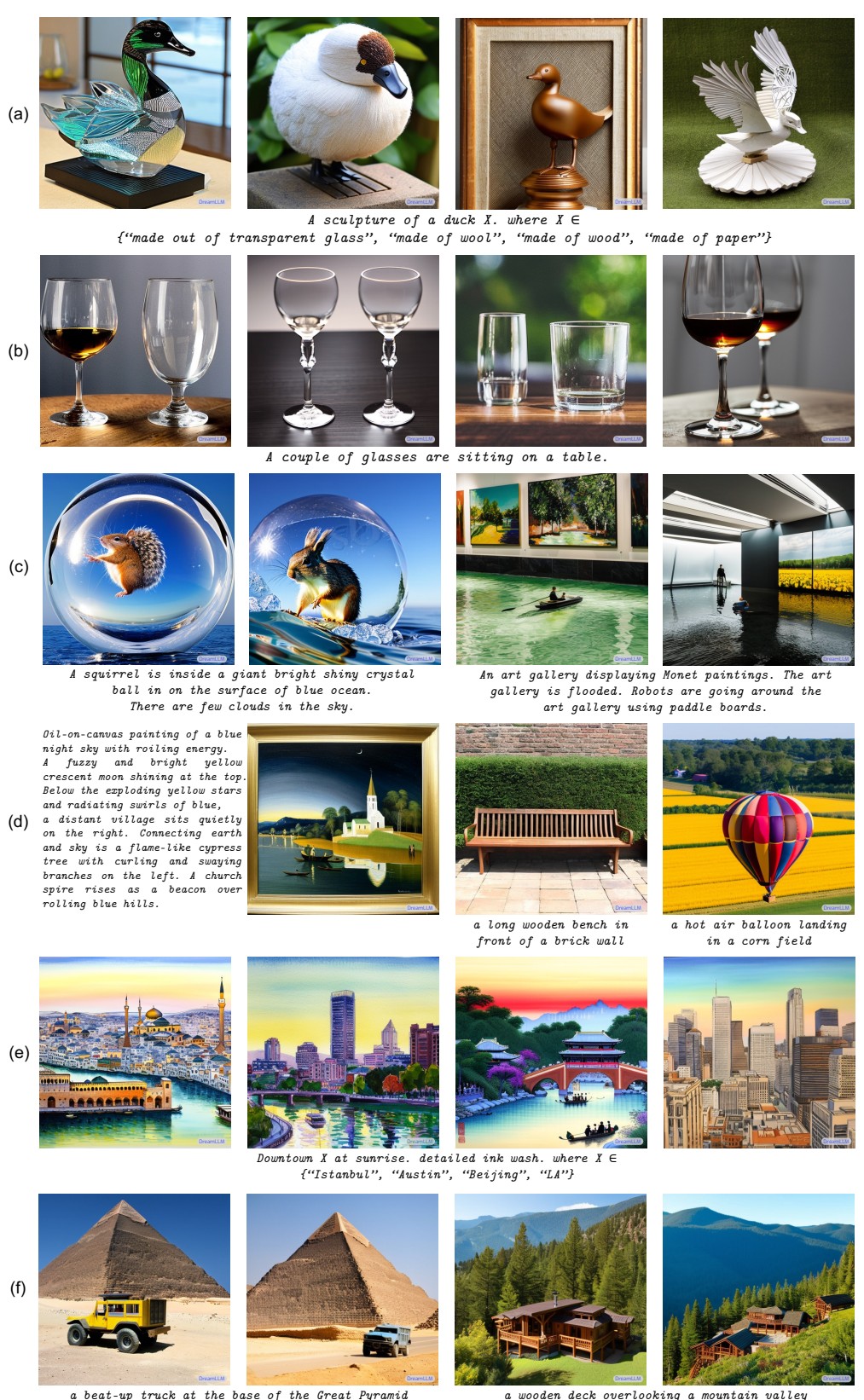

Figure 14: DREAMLLM text-conditional image generation examples with prompts from (a-c) Imagen and DrawBench (Saharia et al., 2022), (d-f) Parti (*i.e.*, PartiPrompts or P2) (Yu et al., 2022b).

