# OpenReview forum: "DreamLLM: Synergistic Multimodal Comprehension and Creation"
_ICLR.cc/2024/Conference — ICLR 2024 spotlight_

### Official Review · Reviewer_jWvU · 2023-10-29

**Soundness:** 3 good
**Presentation:** 4 excellent
**Contribution:** 3 good
**Rating:** 6
**Confidence:** 4

**Summary:**

This paper proposes a framework that unifies generation of text and images. Specifically, it utilizes a newly introduced <dream> token to encode the representations that will later be forwarded to the diffusion model to decode to images. The authors have conducted extensive experiments across multiple tasks and benchmarks to showcase the ability of the proposed method.

**Strengths:**

The paper proposes a unified and promising framework for multimodal generation, strong performances are reported. The proposed approach to integrate diffusion models with LLM seems reasonable and could inspire following works in this area.

**Weaknesses:**

The major concern I have is the necessity to utilize the token from LLM for image decoding. What is is going to be if you let the LLM to first output the image description, then extract it and feed it directly to the diffusion model? In this case, the original text encoder of diffusion models are leveraged. The results in Table 2 show that the specialists still outperform dreamLLM, which means the above naive alternative could potentially perform better? In addition, is it possible that directly use output text can also alleviate the loss of LLM's original power?

**Questions:**

See weakness .

---

> ### Author Response · Authors · 2023-11-21
> **Response to Reviewer jWvU (Part #1)**
>
> Thanks for your positive, insightful reviews and your acknowledgment of our contribution to the community! We respond to your major concerns as follows.
>
> > **Q1:** What is it going to be if you let the LLM first output the image description, then extract it and feed it directly to the diffusion model? ... In addition, is it possible that directly using output text can also alleviate the loss of LLM's original power?
>
> **Analysis of rewrite-then-generate strategy.** Thank you for pointing out this strategy, which has been used by the recently released DALLE-3 [1] based on GPT-4 (released less than a week before the Sep 28'23, ICLR'24 submission deadline). We agree that this `rewrite-then-generate` strategy has the advantage of using textual prompts that may be better suited for specific image synthesis decoders (including DreamLLM). However, there are some challenges and limitations that are non-trivial to overcome:
>
> - **Lacking in data.** The first and the major challenge is the serious lack of training data. In order to train a Large Language Model (LLM) to generate preferred prompts for an image decoder, a dataset specifically tailored to this need is required. For instance, DALLE-3 necessitates the initial training of a bespoke image captioning specialist capable of producing high-quality descriptive captions, followed by model training in a data-rich environment featuring these written captions. This process is far from trivial, hinging heavily on the availability of substantial volumes of high-quality data. However, the interleaved MMC4 dataset lacks alternative versions of rewritten prompts, offering only original internet content. Additionally, different image decoders may perform optimally with different prompts. The collection or labeling of such datasets is a non-trivial, costly endeavor, particularly for the research community.
> - **Limitation in Image-Conditional Image Generation Scalability.** A further limitation lies in the model's inability to scale efficiently in the context of image/document conditional image generation. Given that the image decoder only receives text inputs, no information or details about conditional images can be parsed to serve as conditions for image synthesis. This limitation becomes evident when attempting subject-driven image generation. Please refer to **Appendix A.4, Fig.8, page 25** for our newly conducted **subject-driven image generation** experiments. Moreover, when applied to document-conditional image generation, the text may be excessively lengthy for effective image synthesis. For instance, the maximum token length limitation for the original Stable Diffusion is a mere 77. Furthermore, while prompts generated in context may be suitable for image synthesis, they might appear incongruous when featured in specific documents, thereby disrupting the document's coherency.
> - **LLMs as agents?** We agree that specialist image decoders can be used if we just view the LLMs as agents to call different APIs, but it has its limitations. For example, the data lacking issues and model limitations, as mentioned before, remain. Furthermore, the key limitation may be the **non-end-to-end** pipeline. While tasks can be executed effectively, this disjointed training strategy offers no assurances for potential learning synergies between multimodal comprehension and generation. In principle, this method echoes the Flamingo-like models, which are limited to language-only output. This differs from the **research motivation and purpose** of our work and exploration. Essentially, our exploration of DreamLLM has unveiled the significant potential of LLMs to attain a comprehensive understanding of multimodality that **genuinely comprehends modalities beyond mere language**. It is **novel and presents a promising solution for future-generation MLLMs**, particularly when scaling up LLMs or enhancing the training data (making it more abundant, cleaner, and larger) becomes feasible.

---

> ### Author Response · Authors · 2023-11-21
> **Response to Reviewer jWvU (Part #2)**
>
> **Additional experiments.** Following your instructions, we endeavored to train a model using the proposed `rewrite-then-generate` strategy. Given the absence of an optimal dataset, we modify the original MMC4 by using <dream> start \& end tokens before and after the specific text prompt that has the highest CLIP similarity to a specific image, which can be used as text prompts for image generation. In this setting, we only train the LLMs to output texts, and no image decoders are involved during training. During inference, when the model outputs texts encompassed by the <dream> tokens, the texts are used for an off-the-shelf SD image decoder for generating images. After training, we test the model's language processing and multimodal capabilities.
>
> - **Language processing.** The results in the following table show that the `rewrite-then-generate` achieves similar performance to DreamLLM. This demonstrates that both methods won't impact the language capability, which is as expected.
>
> | Method                  | PIQA     | SIQA     | HellaSwag | WinoGrande | BoolQ    | MMLU     |
> | :---------------------- | -------- | -------- | --------- | ---------- | -------- | -------- |
> | Vicuna Baseline         | 77.7     | 47.5     | 75.7      | 67.5       | 73.9     | 45.0     |
> | DreamLLM                | **78.6** | **48.8** | **77.4**  | **68.5**   | 75.2     | 41.8     |
> | `rewrite-then-generate` | 78.2     | 48.5     | 75.8      | 68.3       | **77.4** | **43.1** |
>
> - **Multimodal comprehension and creation.** The results presented in the subsequent table clearly indicate that the performance of the `rewrite-then-generate` baseline falls short when compared to DreamLLM, particularly in the context of text-to-image generation on the COCO dataset. This underlines the efficacy of the synergistic learning approach inherent in DreamLLM, suggesting its potential superiority over the baseline methodology.
>
> | Method                  | VQAv2 | MM-Vet | COCO  |
> | :---------------------- | ----- | ------ | ----- |
> | DreamLLM-7B             | 56.6  | 35.9   | 8.46  |
> | `rewrite-then-generate` | 54.2  | 34.1   | 11.91 |
>
>
> > **Q2:** The results in Table 2 show that the specialists still outperform DreamLLM, which means the above naive alternative could potentially perform better?
>
> - **Comparison.** The other state-of-the-art text-to-image specialist models such as Imagen-3.4B and Parti-20B have **different image decoder architectures** and are trained on **broad in-house data**. Besides, these models are **not open-sourced**, and thus, we cannot leverage them as our DreamLLM image decoder. However, compared to these SOTA models, our DreamLLM still demonstrates strong performance, successfully improving the SD2.1 baseline to be comparable to these powerful specific models.
> - **Clarification.** We want to clarify that in this work, we mainly build DreamLLM based on **open-sourced SD2.1 baseline**. Compared to the SD2.1 baseline, Table 2 demonstrates that DreamLLM brings **a significant improvement of 3.97/13.73 FID on MS-COCO/LN-COCO, respectively**.
> - **Scalability of DreamLLM.** Our DreamLLM is a general learning framework that does not rely on specific image decoders or LLMs architectures. Therefore, it reveals the great potential and scalability of DreamLLM when incorporating more powerful image decoder baselines or stronger LLMs, if available.
>
> [1] Betker et al. Improving Image Generation with Better Captions. OpenAI.

---

> > ### Comment · Reviewer_jWvU · 2023-11-22
> > **Response to authors**
> >
> > I appreciate the authors' effort in conducting the comprehensive experiments during rebuttal period. I strongly recommend the authors to add those experiments to the main paper in their final version. Given that the authors have addressed most of my concerns, and the paper indeed proposes a novel approach in multimodal learning, I raise my score to leaning acceptance for this paper.

---

> > > ### Author Response · Authors · 2023-11-22
> > > **Thank you for your feedback!**
> > >
> > > Dear Reviewer jWvU,
> > >
> > > Thank you for acknowledging our response and efforts! We're glad to hear that most of your concerns have been addressed.
> > >
> > > We have made a new revision that incorporates the rebuttal experiments in **Appendix A.6, Tab.10, page 26**. Thank you for your insightful suggestions!
> > >
> > > Best Regards,\
> > > Authors

---

### Official Review · Reviewer_t2eT · 2023-11-01

**Soundness:** 3 good
**Presentation:** 3 good
**Contribution:** 3 good
**Rating:** 8
**Confidence:** 4

**Summary:**

This paper proposes a framework to allow multimodal large language models combining multimodal comprehension and generation. To enable image generation, instead of fitting CLIP feature, this work directly optimizes the diffusion model's objective to achieve modeling multimodal posteriors. The training pipeline is comprised of three stages: alignment pretraining, interleaved generative pretraining, and supervised finetuning. Extensive experiments on multimodal comprehension and creation have shown the superiority of the proposed model.

**Strengths:**

1. This model proposed a unified framework for joint multimodal comprehension and generation which shows impressive performance on various tasks, demonstrating the benefits of synergizing these two tasks.
2. The usage of score distillation avoids the possible information loss to greatly improve the image generation ability.
3. The proposed training pipeline enables the free-form interleaved generative ability of multimodal models.
4. The experiments are comprehensive and convincing.

**Weaknesses:**

1. For free-form interleaved generation, it is important to ensure the consistency between related images. However, as we can view the model as replacing the CLIP text encoder of stable diffusion with a much more powerful LLM for the image generation aspect, the control of the generated image is still limited. As we can see from the Figure 3, the phones in generated samples have large discrepancy.
2. The paper does not demonstrate the in-context comprehension ability of the model.
3. Ablation studies on the choices, combination ways, and the importance of filtering process of the training datasets are not shown, which might provide insights for future study.

**Questions:**

Will including samples from training datasets as in-context examples improve the performance during evaluation?

---

> ### Author Response · Authors · 2023-11-21
> **Response to Reviewer t2eT**
>
> Thanks for your solid and positive reviews, which make significant contributions to our manuscript revision, and we are glad that you're pleased with our work! We respond to the Weakness points as follows, which have been incorporated in the revision.
>
> > **W1:** The control of the generated image is still limited considering consistency between images in interleaved generation.
>
> **Clarification.** We highly appreciate your insights into this perceived limitation. In fact, the proposed DreamLLM model architecture is able to generate highly consistent images. However, the images in the interleaved MMC4 dataset used to train DreamLLM inherently have low consistency as it is collected from the noisy Internet. This factor could potentially compromise the model's ability to fully manifest an emergent image consistency property.
>
> **Evidence from subject-driven generation.** To substantiate our assertion that data with superior image consistency can fully leverage the capabilities of DreamLLM in generating consistent images, we have conducted additional experiments focusing on subject-driven image generation. This type of image generation necessitates the model to comprehend intricate instructions while concurrently preserving the subject identity features with utmost precision. Our DreamLLM has been fine-tuned for image-to-image generation in the third stage of SFT using the same data employed by BLIP-Diffusion [1], a recently proposed model in the domain of subject-driven generation. As this data set is not publicly available, we have constructed it independently based on the procedures outlined in the original paper. The results are shown in **Appendix A.4, Fig.8, page 25**. The results compellingly validate the effectiveness and great potential of applying DreamLLM for image-consistent generation.
>
> [1] Li et al. BLIP-Diffusion: Pre-trained Subject Representation for Controllable Text-to-Image Generation and Editing. In NeurIPS 2023.
>
> > **W2:** Will including samples from training datasets as in-context examples improve the performance during evaluation?
>
> Indeed. We extend our gratitude for your insightful suggestion! Please refer to our [General Response](https://openreview.net/forum?id=y01KGvd9Bw&noteId=xe4TvtyFfS) for the in-context learning and few-shot experimental results, the results demonstrate the powerful in-context learning and few-shot learning capabilities of DreamLLM.
>
> > **W3:** Ablation studies on training datasets are not shown.
>
> Thanks for your suggestions! Due to time limitations, it is impractical to perform a comprehensive ablation study encompassing all training datasets and filtering strategies, as such an undertaking may indeed exceed the scope of the current research. However, during rebuttal, we have conducted some ablation studies on instruction-following data and training data. We kindly refer you to our response to **Q1.a of Reviewer h8i6** for details. In addition, please note that the filtering strategies on datasets such as MMC4 are basically intuitive. During our manual review of the data, we identified several instances of misalignment between images and their corresponding text, which served as an impetus for our filtering approach. It is our future work to identify and resolve all data-related discrepancies.

---

> > ### Comment · Reviewer_t2eT · 2023-11-22
> > **Responce to Authors**
> >
> > Thanks for your feedback and additional results. I would like to keep my original rating.

---

> ### Author Response · Authors · 2023-11-23
> **Thank you!**
>
> Dear Reviewer t2eT,
>
> We sincerely appreciate your positive review and acknowledgment of our response!
>
> Best Regards,\
> Authors

---

### Official Review · Reviewer_h8i6 · 2023-11-01

**Soundness:** 2 fair
**Presentation:** 3 good
**Contribution:** 3 good
**Rating:** 6
**Confidence:** 4

**Summary:**

The paper introduces "DREAM LLM", a novel learning framework designed to enhance Multimodal Large Language Models (MLLMs). The model is designed to 1) The model late-fused three pretrained models, a text-LLM, an image generation model (SD), a CLIP model. Unlike conventional methods that utilize intermediate image representations, DREAM LLM leverages score distillation techniques with a diffusion image generation model inputs raw data modalities and outputs them in the same format. 2) The model is also trained on interleaved multimodal corpora sourced from the internet.
This leads to superior performance in several benchmarks and the ability to generate free-form interleaved content.

**Strengths:**

1. The model late-fused three pretrained models, a text-LLM, an image generation model (SD), a CLIP model. With the designed dream query and score distillation with SD model, the model is superior than earlier models that utilize intermediate image representation.
2. What's more, the model is trained on carefully collected various datasets, including, various text-image datasets, interleaved multimodal corpora sourced from the internet, instruction datasets. With joint-training, it also shows synergy of text and image, understanding and creation.
3. The model shows very comprehensive experiment results in various benchmarks like MS-COCO, MMBench, and MM-Vet, in different setup zero-shot understanding, image generation, interleaved generation, etc.

**Weaknesses:**

1. The model is trained on very rich data, and it is not clear how does those data contribute to the zero-shot evaluation.

a. several dataset used by the model are derived from COCO datasets, e.g Laion-COCO, LLaVaInstruct, etc. How do we know if there is data leakage in the training datasets. This applies to the results in table 1, as well as in table2.

b. Is the model in table 1 after instruction tuning?

c. if you continue training SDv2.1 with the collected dataset, what are the MS-COCO, and LN-COCO FID number?

2. Some model details are not clear.

a. how is the multi token dream query implemented. For decoder-only transformer training, every token needs a loss (or a score). What us the loss of each query token during training, and are they generated sequentially or altogether during inference?

b. in the interleaved multimodal joint training, for the second/third image generation, do they condition on both the image1 dream query and image1 visual encoder? Or just image1 visual encoder.

c. for the stage1, stage2, stage3, what's the final loss? Do they both have L_DM (formula 5) and L_MLLM (formula 6)? Is there a weight?

d. for the I-GPT training, does the visual projector, condition projector, dream embedding get updated? or only the MLLM transformer got updated?

e. in section 5.1, the L_CLIP is not clear. Which two embeddings are used to calculate the loss?

**Questions:**

1. Can the model do k-shot learning for image understanding task?
2. table4 shows amazing results that the model is better at text task than the pretrained text LLM, does the author have any hypothesis why?

---

> ### Author Response · Authors · 2023-11-21
> **Response to Reviewer h8i6 (Part #1)**
>
> Thank you for your constructive and supportive reviews that greatly improve our manuscript! We respond below to your questions and concerns, which we have incorporated into the revised version.
>
> > **Q1.a:** Is there data leakage?
>
> | Stage | Objective     | Trainable                              |        Dataset        |   Image Source    | Data Size |                          Reference                           |
> | ----- | :------------ | -------------------------------------- | :-------------------: | :---------------: | :-------: | :----------------------------------------------------------: |
> | I     | Comprehension | visual projector                       |    `LLaVAPretrain`    |      `CC-3M`      |   558K    | [URL](https://huggingface.co/datasets/liuhaotian/LLaVA-CC3M-Pretrain-595K) |
> |       | Comprehension | visual projector                       |      `LAION400M`      |    `LAION400M`    |    11M    |     [URL](https://laion.ai/blog/laion-400-open-dataset/)     |
> |       | Creation      | condition projector + dream embeddings |      `LAION400M`      |    `LAION400M`    |    11M    |     [URL](https://laion.ai/blog/laion-400-open-dataset/)     |
> |       | Creation      | condition projector + dream embeddings |     `BLIP-LAION`      |    `LAION400M`    |    8M     | [URL](https://github.com/salesforce/BLIP#pre-training-datasets-download) |
> |       | Creation      | condition projector + dream embeddings |     `LIAON-COCO`      |   `LAION2B-EN`    |    11M    |           [URL](https://laion.ai/blog/laion-coco/)           |
> | II    | Both          | ALL (including LLM)                    |        `MMC4`         |      `MMC4`       |    2M     |            [URL](https://github.com/allenai/mmc4)            |
> |       | Creation      | ALL (including LLM)                    |     `BLIP-LAION`      |    `LAION400M`    |    8M     | [URL](https://github.com/salesforce/BLIP#pre-training-datasets-download) |
> | III   | Both          | ALL (including LLM)                    |    `InstructMMC4`     |      `MMC4`       |    20K    |                             N/A                              |
> |       | Creation      | ALL (including LLM)                    | `Instruct-BLIP-LAION` |    `LAION400M`    |    20K    |                             N/A                              |
> |       | Comprehension | ALL (including LLM)                    |    `LLaVAInstruct`    | `COCO train 2017` |    80K    | [URL](https://github.com/haotian-liu/LLaVA/blob/main/docs/Data.md) |
>
> To answer this question, we revisit the datasets used in each training stage, as detailed in the previous table, to enhance comprehension of training data utilization.
>
> **Comprehension. (Tab. 1)** The provided table clearly demonstrates that only `LLaVAInstruct`, utilized during Stage III SFT training, incorporates images from `COCO train 2017`. However, several key points should be noted:
>
> - Only images from the `train` split are used, which have no overlap with the images used in evaluations. This means **no images in the evaluation benchmarks are leaked**.
>
> - `LLaVAInstruct` is constructed by LLaVA using GPT-4. It's **different from any of the training/validation data used in our evaluations like VQAv2**. It's worth noting that LLaVA and Emu, which we're comparing, also use this `LLaVAInstruct` dataset during SFT.
>
> - Following the reviewers' suggestions on a further ablation study of the effect of `LLaVAInstruct` during Stage III SFT training, we conduct Stage III SFT by replacing the instruction-tuning comprehension dataset `LLaVAInstruct` with an instruction-following data that truly covers some VQA training datasets in our evaluations (other SFT datasets are the same). To this end, we use `LLaVAInstruct 1.5`, which is a new visual instruction-following dataset constructed by LLaVA 1.5 [1] very recently (post Sep 28'23, ICLR'24 submission deadline). This is collected by mixing a variety of QA, VQA, conversation, and other type datasets into instruction-following conversations, including VQAv2 and OKVQA, that are used in our evaluations. Under this setting, the results will not be zero-shot for VQA on VQAv2 and OKVQA.
>
>   The ensuing table presents the results of our study. These results compellingly illustrate that for both VQAv2 and OKVQA, the application of the training dataset leads to a substantial enhancement in accuracy. Besides, it is noteworthy that a remarkable improvement is achieved on datasets like VizWiz that consist of substantially different images from COCO (i.e., images taken by blind people with the VizWiz software).
>
>   | Instruct Dataset           | VQAv2 | VizWiz | OKVQA | TextVQA | MM-Vet |
>   | -------------------------- | :---: | ------ | :---: | :-----: | :----: |
>   | `LLaVAInstruct` (80K)      | 56.6  | 45.8   | 44.3  |  34.9   |  35.9  |
>   | `LLaVAInstruct 1.5` (665K) | 72.9  | 49.3   | 53.3  |  41.8   |  36.6  |
>   | $\Delta$ Acc.              | +16.3 | +3.5  | +9.0  |  +6.9   |  +1.0  |

---

> > ### Author Response · Authors · 2023-11-21
> > **Response to Reviewer h8i6 (Part #2)**
> >
> > **Creation. (Tab. 2)** There is **no** data leakage for the evaluation. **No images from COCO are used as creation targets during training.** We want to emphasize that `LAION-COCO` or `BLIP-LAION` images are from the LAION dataset, but it is constructed by replacing the original noisy Internet captions with captions made by COCO-pretrained model BLIP. To verify if the COCO-style captions benefit downstream COCO FID, we conduct experiments by fine-tuning SDv2.1 on the same training dataset as DreamLLM. The results are shown in the following table. Note that SDv2.1 is pretrained on LAION-5B with unknown details, and all models below are based on a pretrained SDv2.1. From the results, it can be observed that the training data used by DreamLLM indeed benefits SDv2.1 on COCO text-to-image FID. This may come from the COCO-style captions, which are close to the COCO dataset. However, our DreamLLM still outperforms the SDv2.1 baseline by a clear margin.
> >
> > | Model              | Data                                                       | MS-COCO  | LN-COCO   |
> > | ------------------ | ---------------------------------------------------------- | -------- | --------- |
> > | SDv2.1             | Unknown Details                                            | 12.43    | 34.26     |
> > | SDv2.1             | `LAION400M` (11M) + `BLIP-LAION` (8M) + `LAION-COCO` (11M) | 11.91    | 25.35     |
> > | DreamLLM-7B (Ours) | `LAION400M` (11M) + `BLIP-LAION` (8M) + `LAION-COCO` (11M) | **8.46** | **20.53** |
> >
> > > **Q1.b:**  Is the model in table 1 after instruction tuning?
> >
> > Yes. Only after undergoing instruction tuning is the model sufficiently prepared to function as a multimodal generalist. This enables it to execute complex dialogues or engage in creative tasks such as the generation of interleaved content guided by specific instructions.
> >
> > > **Q1.c:** If you continue training SDv2.1 with the collected dataset, what are the MS-COCO, and LN-COCO FID number?
> >
> > Following your suggestion, we fine-tuned SD2.1 with our training data, and the results are shown in our response to question **Q1.a**. From the results, it can be observed that the data used for training DreamLLM indeed improves SDv2.1 on COCO text-to-image FID. However, DreamLLM still outperforms SDv2.1 by a clear margin.  Besides, we want to note that the training data is mixed because we are only able to download partial subsets of each dataset due to the unstable Internet connection. Hence, we just mix the datasets to approximately 30M intuitively, there is no careful selection of datasets.

---

> > > ### Author Response · Authors · 2023-11-21
> > > **Response to Reviewer h8i6 (Part #3)**
> > >
> > > > **Q2.a:** How is the multi token dream query implemented? For decoder-only transformer training, every token needs a loss (or a score). What is the loss of each query token during training, and are they generated sequentially or altogether during inference?
> > >
> > > - The multi-token dream query is implemented as learnable parameters, i.e., `nn.Parameter()`.
> > > - During training, only the text tokens (including some special tokens such as the \<dream\> tokens) need a per-token language modeling loss (i.e., Cross-Entropy loss). The query tokens are fed **altogether** into the SD image decoder, and it is supervised by directly optimizing the **diffusion loss** (i.e., noise prediction MSE loss).
> > > - During inference, the pretrained dream queries are fed altogether into the LLM when the special \<dream\> token is generated. The dream queries do not need to be generated since they are learnable parameters that are part of DreamLLM.
> > >
> > > > **Q2.c:** For the stage1, stage2, stage3, what's the final loss? Do they both have L\_DM (formula 5) and L\_MLLM (formula 6)? Is there a weight?
> > >
> > > Sorry for the confusion about the training details. The detailed training implementation is listed in **the first table** in our response to **Q1.a**. $\mathcal{L}\_{\text{DM}}$ is used for training `creation` , while $\mathcal{L}\_{\text{MLLM}}$ is used for training `comprehension`. If the objective is `both`, then we sum the two losses with a weight $\alpha$: $\mathcal{L}\_{\text{MLLM}} + \alpha\mathcal{L}\_{\text{DM}}$. We set $\alpha=10$ in this work to balance the loss.
> > >
> > > > **Q2.d:** For the I-GPT training, does the visual projector, condition projector, dream embedding get updated? or only the MLLM transformer got updated?
> > >
> > > Sorry for the confusion about the training details. During $\mathcal{I}$-GPT training, all model components are trained jointly, including the visual projector, condition projector, dream embeddings, and the LLM. We have presented a more detailed training configuration in **the first table** in our response to **Q1.a**.
> > >
> > > > **Q2.e:** In section 5.1, the L_CLIP is not clear. Which two embeddings are used to calculate the loss?
> > >
> > > Sorry for the confusion. This is a typo. The $\mathcal{L}\_{\text{CLIP}}$ should be $\mathcal{L}\_{\text{align}}$, which is defined and discussed in **Section 2.1**. The definition is $\mathcal{L}\_{\text{align}}=D(\mathcal{M}\_\psi\circ\mathcal{C}^{\text{MLLM}}, \mathcal{C}^{\text{CLIP}})$, which is a **per-token feature alignment loss (e.g., MSE loss used by Emu) between the LLM-projected features and CLIP features**. However, our DreamLLM does **not** rely on such CLIP alignment, and the analysis is conducted in **Section 5.1**.
> > >
> > > > **Q3:** Can the model do k-shot learning for image understanding task?
> > >
> > > Thank you for your constructive suggestion!  Please refer to our [General Response](https://openreview.net/forum?id=y01KGvd9Bw&noteId=xe4TvtyFfS) for the in-context learning and few-shot experimental results, the results demonstrate the powerful in-context learning and few-shot learning capabilities of DreamLLM.

---

> ### Author Response · Authors · 2023-11-23
> **Looking forward to your feedback**
>
> Dear Reviewer h8i6,
>
> Thank you again for your time and constructive reviews! With the discussion period drawing to a close **in several hours**, we expect your feedback and thoughts on our reply. We put a significant effort into our response, with several new experiments and discussions. We sincerely hope you can consider our reply in your assessment. We look forward to hearing from you, and we can further address unclear explanations and remaining concerns, if any.
>
> Best Regards,\
> Authors

---

> > ### Comment · Reviewer_h8i6 · 2023-11-23
> > **Responce to Authors**
> >
> > Thanks for your feedback and additional results. I would like to keep my original rating.

---

> > > ### Author Response · Authors · 2023-11-23
> > > **Thank you!**
> > >
> > > Thank you for your acknowledgment! We are glad that you are satisfied.

---

### Author Response · Authors · 2023-11-21
**General Response**

*We sincerely appreciate all reviewers and community members for their efforts in evaluating the paper and writing suggestions that greatly help us improve the work. Here, we respond to a common concern from **Reviewer h8i6** and **Reviewer t2eT**. We have revised the manuscript accordingly, and the revision details are listed here.*

### Few-Shot Comprehension
The table below presents our $k$-shot comprehension results. Following Flamingo and Emu, we utilize the RICES technique [1] for sample selection by identifying the most pertinent examples. The results have been incorporated in **Appendix A.3, Tab. 8, page 24**. The results clearly underscore the superior in-context learning power of DreamLLM in comparison to Emu and Flamingo. These results provide compelling evidence of DreamLLM's proficiency in harnessing and utilizing in-context knowledge.

| Model                           | VQAv2 $k$-shot |      |      |      | VizWiz $k$-shot |      |      |
| :------------------------------ | -------------- | ---- | ---- | ---- | --------------- | ---- | ---- |
| $k$                             | 2              | 4    | 8    |      | 2               | 4    | 8    |
| Kosmos-1                        | 51.4           | 51.8 | 51.4 |      | 31.4            | 35.3 | 39.0 |
| Flamingo-9B                     | -              | 56.3 | 58.0 |      | -               | 34.9 | 39.4 |
| Emu-14B                         | 56.4           | 58.4 | 59.0 |      | 37.8            | 41.3 | 43.9 |
| DreamLLM-7B                     | 58.1           | 59.2 | 59.4 |      | 46.1            | 46.7 | 46.8 |
| DreamLLM-7B (LLaVAInstruct 1.5) | 73.8           | 74.4 | 73.8 |      | 49.8            | 50.3 | 49.7 |

[1] Yang et al. An Empirical Study of GPT-3 for Few-Shot Knowledge-Based VQA. In AAAI 2022.

### In-Context Comprehension

In-context comprehension is a critical capability for a foundation model. Please refer to **Appendix A.3, Fig. 7, page 24** for examples of this in DreamLLM, exhibiting proficiency akin to prior models such as Flamingo and Emu. Note that compared to in-context image generation, in-context comprehension may be simpler as the hints from texts are generally denser, which could help in-context comprehension. However, in-context generation requires a more comprehensive understanding of image details and text semantics for preserving subject features and performing reasoning.

### Paper Revision

Following all reviewers' constructive instructions, we have carefully incorporated the rebuttal experiments to revise the paper. All revised texts are colored blue. We summarize the revision as follows:

1. We add the few-shot multimodal comprehension experiments in Appendix A.3.
2. We add the in-context comprehension qualitative examples in Appendix A.3.
3. We add a subject-driven image generation experiment in Appendix A.4.
4. We add the discussion and comparison experiment to rewrite-then-generate baseline in Appendix A.6.
5. We have carefully revised the paper to fix all typos.

### Looking forward to your feedback
With the discussion period drawing to a close, we expect your feedback and thoughts on our reply. We put a significant effort into our response, with several new experiments and discussions. We sincerely hope you can consider our reply in your assessment. We look forward to hearing from you, and we can further address unclear explanations and remaining concerns, if any.

---

### Meta-Review · Area_Chair_JnZj · 2023-12-15

**Metareview:**

This submission introduces DreamLLM, a new framework that unifies text and image generation in multimodal Large Language Models (MLLMs). Employing a novel <dream> token and score distillation with a diffusion model, the paper stands out for its novel approach to handling raw data modalities. The experiments across diverse benchmarks effectively demonstrates the model's superiority in both multimodal comprehension and generation.

The reviewers initially raised concerns regarding aspects such as the necessity of the <dream> token, control over generated images, and the model's in-context comprehension abilities. However, the authors' comprehensive and detailed responses, coupled with additional experiments, addressed most of these issues, leading to increased confidence and positive ratings from the reviewers. The most notable contribution of this submission is its pioneering approach in the seamless integration of different modalities within a single model framework, which not only enhances the understanding and generation capabilities of MLLMs but also paves the way for future advancements in this rapidly evolving field. The AC recommends accepting this paper for publication.

**Justification For Why Not Higher Score:**

While this paper is a strong and commendable piece of work deserving of its spotlight status, the model's in-context learning abilities are not fully justified (which however is not a flaw IMO and the authors shouldn't take the justification for not higher/lower scores too seriously).

**Justification For Why Not Lower Score:**

This work marks a significant leap in the realm of MLLMs, showcasing a unique and effective integration of text and image modalities.

---

### Decision · Program_Chairs · 2024-01-16

Accept (spotlight)